# A Mississippian (early Carboniferous) tetrapod showing early diversification of the hindlimbs

Jennifer A. Clack[1,3], Timothy R. Smithson [1✉] & Marcello Ruta [2]

The taxonomically diverse terrestrial tetrapod fauna from the late Mississippian East Kirkton Limestone includes the earliest known members of stem Amphibia and stem Amniota. Here we name and describe a new East Kirkton tetrapod with an unusual hindlimb morphology reminiscent of that of several stem- and primitive crown amniotes. It displays a unique ilium with two slender and elongate processes and a 5-digit pes with a long, stout metatarsal IV and a greatly elongate digit IV. The new taxon broadens our knowledge of East Kirkton tetrapods, adding to the remarkable diversity of their hindlimb constructions, functional specializations, locomotory modes, and adaptations to a wide variety of substrates. An unweighted character parsimony analysis places the new taxon in a polytomy alongside some other Carboniferous groups. Conversely, weighted parsimony and Bayesian analyses retrieve it among the earliest diverging stem amniotes, either as the basalmost anthracosaur or within a clade that includes also *Eldeceeon* and *Silvanerpeton*, crownward of an array of chroniosaurs plus anthracosaurs.

[1] University Museum of Zoology Cambridge, Downing Street, Cambridge CB2 3EJ, UK. [2] School of Life and Environmental Sciences, University of Lincoln, Joseph Banks Laboratories, Green Lane, Lincoln LN6 7DL, UK. [3] Deceased: Jennifer A. Clack. ✉email: ts556@cam.ac.uk

The Carboniferous tetrapod fauna from the East Kirkton Limestone of the Bathgate Hills in Scotland provides a unique window onto the diversity of terrestrial vertebrates during the late Mississippian (early Carboniferous), ~336 Ma (million years ago). The seven East Kirkton tetrapods described and named so far are both taxonomically diverse and morphologically disparate, exhibiting a wide range of body shapes and sizes, vertebral constructions, and limb proportions. This level of diversity among tetrapods is not encountered again until the mid-Pennsylvanian, some 25 million years (Myr) later, by which time several clades, particularly among crown Amniota, are well established and greatly diversified[1]. Among the tetrapods represented at East Kirkton are the temnospondyl *Balanerpeton*[2], the anthracosaur-like *Silvanerpeton*[3,4] and *Eldeceeon*[5,6], and the amniote-like *Westlothiana*[7]. In some recent studies[8–10], these taxa have been placed phylogenetically as the earliest known members of stem Amphibia (*Balanerpeton*) or stem Amniota (*Westlothiana*; *Silvanerpeton*; *Eldeceeon*; but see ref. [11] for alternative hypotheses on the affinities of many of these taxa, e.g., as stem tetrapods), and thus provide a minimum age estimate for the origin of crown Tetrapoda[8,12]. Other East Kirkton taxa include the baphetid-like *Eucritta*[12,13], the aïstopod *Ophiderpeton*[14], and the putative microsaur *Kirktonecta*[15]. Except for *Ophiderpeton*, all these taxa are among the earliest diverging members of clades that are more commonly found throughout the later Palaeozoic and combine several of the most plesiomorphic traits of those clades.

In this article, we name and describe a new tetrapod from East Kirkton, based upon postcranial remains that show unusual specializations to the pelvic and hindlimb skeletons, until now observed only in many later tetrapods, as well as a suite of unique characteristics. The holotype was collected by the late Stan Wood and donated to the University Museum of Zoology, Cambridge (UMZC), probably in the 1990s, although details of its collection were, unusually, not recorded by him. Possible reasons for this are that the specimen derived either from one of the spoil heaps from the old quarry workings of the locality or from one of the boundary walls that Stan had bought and collected from before the main East Kirkton quarry was discovered[16]. The matrix lithology suggests that the specimen may have originated from Black Shale Unit 82[17], a particularly fossil-rich horizon that has yielded several compressed, but often near-complete and slightly disarticulated tetrapods. The unusual circumstances of fossil preservation, in a locality affected by volcanic activity and including a mineral-rich lake fed by warm or hot springs, have been described in a series of papers in Rolfe et al.[18] and summarized in Clack[19]. As well as the earliest known terrestrial tetrapods, the East Kirkton quarry has also yielded fish, arthropods, and plants[18].

The new taxon prompts a reconsideration of the importance of the East Kirkton site for our understanding of the evolutionary and ecological contexts in which tetrapods radiated and diversified during the Viséan stage of the early Carboniferous. More broadly, its significance lies in the fact that it casts new light on the emergence of tetrapod terrestrial adaptations, such as are revealed by its unusual pelvic and hindlimb skeletons.

## Results
### Systematic palaeontology.

> Tetrapoda Jaekel, 1909 *fide* Sues[20]
> Family undesignated
> *Termonerpeton makrydactylus* gen. et sp. nov. (Fig. 1)

**Etymology**. *Genus*: from τέρμων (térmon) meaning boundary and ερπετό (erpetó) meaning 'crawler', referring to the field boundary walls near the East Kirkton quarry where the late Stan Wood initially discovered fossils from the East Kirkton Limestone and from where the type specimen may have been collected; *species*: from μακρύς (makrýs) meaning 'elongate' and δάχτυλο (dáchtylo; more precisely, δάχτυλο ποδιού, dáchtylo podioú) meaning 'toe', referring to the very long pedal digit IV.

**Holotype**. University of Cambridge Museum of Zoology (UMZC) 2019.1. A partial tetrapod postcranium, preserving both pelves, a femur, fibula, tibia, and an almost complete but disarticulated pes. Closely associated with the appendicular elements are dorsally open hoop-shaped centra, a few neural and haemal arches, curved ribs, and a section of articulated gastralia.

**Locality and horizon**. East Kirkton quarry, near Bathgate, Scotland, UK. East Kirkton Limestone, Bathgate Hills Volcanic Formation. Exact horizon is unknown. Brigantian, Viséan, early Carboniferous (=Mississippian)[21].

**Differential diagnosis**. Possible autapomorphies: ilium with drawn-out, flat, blade-like dorsal process; very large, stout, and elongate metatarsal IV, greatly exceeding the length of metatarsals III and V (~30% or more). Possible tetrapod synapomorphies among post-Devonian taxa: distinct interepipodial space between tibia and fibula; well-ossified tarsus comprising tibiale, fibulare, intermedium, four centralia, and five distal tarsals. Possible amniote synapomorphies, but often showing reversed polarity in several stem- and crown amniote taxa: presumed pedal phalangeal formula 23454; robust and long pedal digit IV; enlarged intermedium and fibulare, together occupying more than half of proximal moiety of tarsus; curved ribs. Characters of uncertain polarity (also present in *Caerorhachis*): elongate, slender, and posterodorsally oblique post-iliac process; short puboischiadic plate with almost vertical anterior margin; stout femur with poorly pronounced waisting along the shaft, longer than puboischiadic plate; hoop-shaped centra.

**Attributed specimen**. National Museums Scotland (NMS) G.1992.22.1. An articulated, partially complete, large tetrapod pes, preserving a nearly complete array of tarsals, all metatarsals, and the proximal phalanges of digits I–III. Unit 82, East Kirkton Limestone, East Kirkton quarry, near Bathgate, Scotland, UK.

Specimen description

**Appendicular skeleton**. Most of the description is based upon the holotype. Both pelves are preserved, one mainly as a natural mould. The puboichiadic plates are short and deep, with an almost vertical anterior margin to the pubis (Fig. 1). In one, the surface of the puboischiadic plate is strongly convex, in the other it is strongly concave. The concave plate may belong to the left pelvis, with the concavity indicating the acetabulum. Both iliac processes of the presumed right ilium are overlain by a neural arch and part of the femur and cannot be seen. The presumed left ilium shows a long, posteriorly pointing post-iliac process that extends as far backward as the posterior edge of the ischium. It retains the proximal, stump-like portion of a dorsal iliac process, continued distally in natural mould as a mediolaterally flattened and blade-like structure. Both processes sit above a short iliac neck. The dorsal iliac process is proportionally longer than in other tetrapods and its knife-like appearance is unique. The angle between the two processes is much more acute than in most other tetrapods, and the nearest comparison is with the divided iliac process of the microsaurs *Tuditanus* and *Ricnodon*[22] which, however, could merely represent a bifid post-iliac process. Two

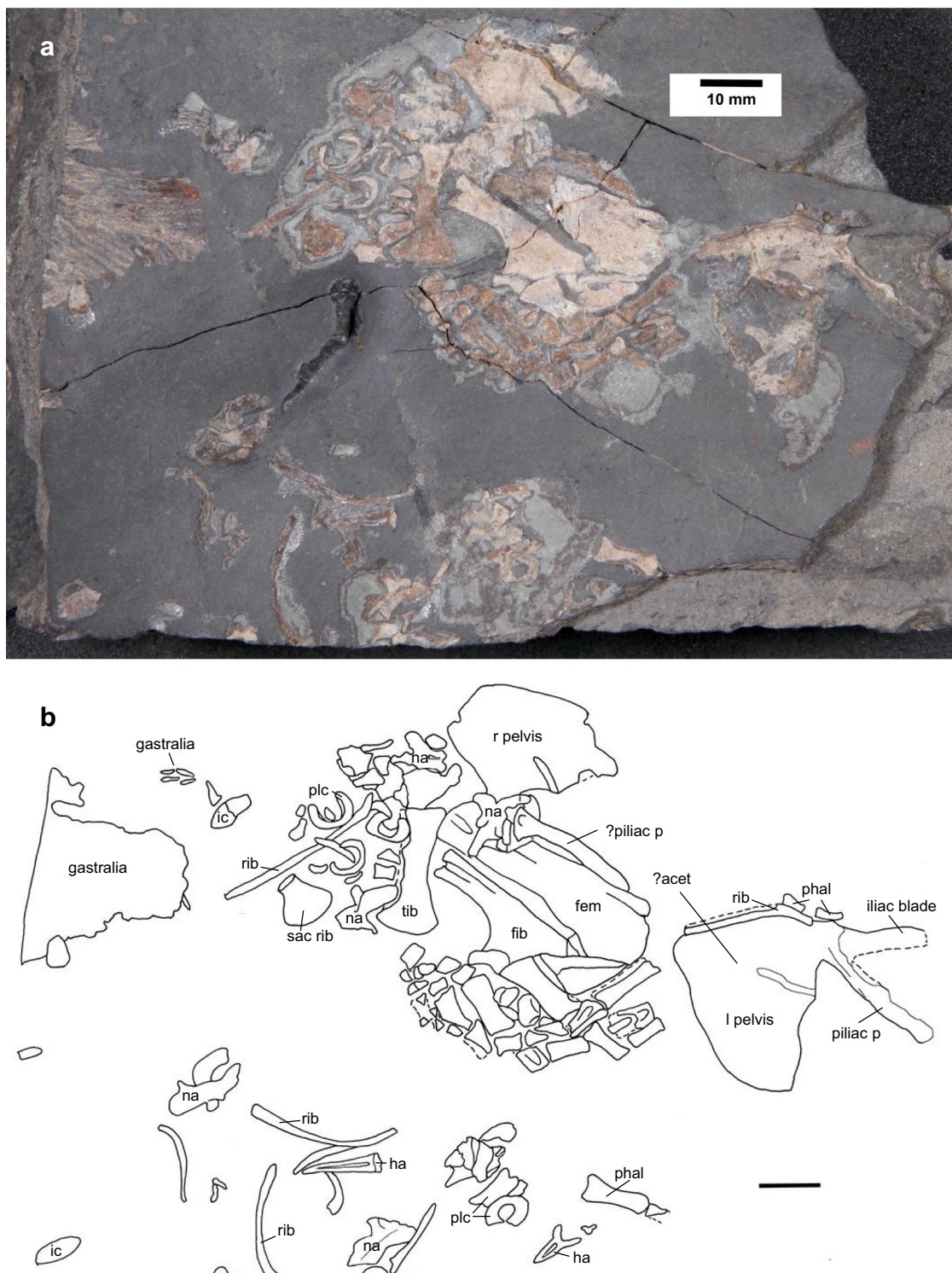

**Fig. 1 *Termonerpeton makrydactylus* gen. et sp. nov. holotype UMZC 2019.1. a** Specimen photograph. **b** Interpretive drawing. Scale bars 10 mm. Abbreviations: acet acetabulum, fem femur, fib fibula, ha haemal arch, ic intercentrum, l left, na neural arch, phal phalanx, piliac p post-iliac process, plc pleurocentrum, r right, sac rib sacral rib, tib tibia.

gaps in ossification are taken as evidence of an ilio-ischiadic suture half-way down the posterior margin on the left pelvis and an ilio-pubic suture halfway down the anterior margin of the right pelvis (Fig. 1). There is no evidence of a puboischiadic suture, although a shallow depression along the ventral margin of the left puboischiadic plate probably marks the junction between pubis and ischium. The complete left puboischiadic plate is 20 mm deep behind the ilium and 30 mm long, with the pubis contributing about one-third of its length and the ischium the remaining two thirds. The anterior margin of the pubis is almost vertical. The dorsal margin of the ischium is shallowly convex for half its length before extending posteroventrally to meet the

upturned posterior extremity of the ischium's ventral margin. There is no evidence as to the angle at which the two pelvic plates met at the symphysis, which would affect the position of the acetabulum relative to the substrate, and thus the effective resting posture of the hindlimb.

The left femur is at least 39 mm in length, and longer than the puboischiadic plate. The entire bone is crushed, and its distal end lies partly beneath one of the pelvic halves and a neural arch so that its features cannot easily be made out. A possible intercondylar groove may be present distally, and the extensor surface of its proximal extremity appears to show a subcentral depression. The femur itself is robust with little waisting at mid-shaft. A small internal trochanter lies near its proximal end. The left fibula is approximately 26 mm long along its lateral margin. Its proximal end is narrow and grooved. Its broad and strongly flared distal end suggests a broad contact with the tarsus. The medial turn of the distal end indicates a large interepipodial space. The left tibia is about 20 mm long, slender, and shallowly waisted at mid-shaft. It is not clear which end is proximal and which distal, although probably the proximal is the broader. The tibia is probably more than half the length of the femur. Based upon the femur and tibia lengths, and omitting the ankle and pes, the above figures indicate a total stylopod-zeugopod length of about 65 mm, assuming a fully extended limb.

Most of the morphology of the left pes is preserved, showing many well-ossified tarsal bones (Fig. 2). Several of these, including possible distal tarsals II and III lie more or less in anatomical continuity relative to metatarsals II and III, respectively. Other tarsal elements, including possible fibulare, tibiale, centralia, and distal tarsals, are illustrated in Fig. 2. Metatarsal IV lies in anatomical position relative to metatarsals II and III and, at 7 mm in length, is significantly larger than the latter. The presumed first phalanx of pedal digit IV lies close to metatarsal IV, at an angle of nearly 90° to the latter. It is long and slender, indicating an unusually elongate fourth pedal digit.

An array of about 12 phalanges is preserved. They are all disrupted but occur in proximity to one another and, like the first phalanx of pedal digit IV, also mainly lie at right angles to metatarsals III and IV. An additional, acutely angled pointed ungual phalanx, possibly associated with digit II, is also visible. A further two phalanges have been displaced and rest along the anterior edge of the left pelvis. In total, we were, therefore, able to identify 15 elements. The preservation of the pes suggests it was strongly flexed either at death or from tissue shrinkage thereafter. An isolated metatarsal, presumably from the other, missing foot, lies some distance away near the edge of the block. Together, the pedal elements suggest a relatively large foot.

A second specimen, NMS G.1992.22.1 (Fig. 3), is represented by an isolated pes. It may belong to *Termonerpeton*, although it is from a much larger individual. It shows five metatarsals of which the fourth is much longer and more robust than the other four and about twice as long as that of the holotype, while metatarsal V is the smallest. There are three phalanges, plus five distal tarsals. A D-shaped element closely associated with three centralia could be either a fibulare, a displaced intermedium, or centrale IV.

**Axial skeleton**. Where visible, neural arches have short neural spines and prominent zygapophyses, but their shape is hard to assess as none is well preserved. The element overlying part of the right pelvis and the femur is 7 mm high in total. Numerous dorsally open, hoop-shaped centra about 5 mm in diameter are visible, as well as a few small, oval, shallowly curved elements (Fig. 1). Without further evidence, it is uncertain which of these elements are intercentra and which pleurocentra, though we assume that the larger elements are pleurocentra. The preserved ribs are slender and curved, and include trunk ribs, a possible presacral rib, a possible sacral rib, and a possible postsacral rib. This is long but more or less straight. A bone situated among a cluster of centra, somewhat distant from the other tarsal bones, was originally interpreted by us as a possible fibulare, similar to the fibulare in *Proterogyrinus*[23]. However, it might also be interpreted as a sacral rib. If so, its morphology is unique. It is short and widens distally into a fan-shaped structure but does not appear to have a bifid proximal end, unlike the sacral rib in *Proterogyrinus*[23]. Three haemal arches are present, one still attached to its half-hoop centrum, a second slightly longer, and a third very short and presumably from a more posterior region of the tail.

**Comparisons**. The exceptional preservation of tetrapods from the East Kirkton Limestone provides a unique opportunity to study portions of the skeletal anatomy that are otherwise poorly preserved or absent among Mississippian tetrapods. In particular, hindlimbs with a complete or near-complete array of tarsal elements and digits are notably rare. The unusual construction of the pes of *Termonerpeton* prompted us to examine the hindlimb morphology of six other East Kirkton tetrapods (Fig. 4a–g) alongside a selection of additional, mostly Carboniferous taxa (Fig. 4h–n). We focus on epipodials, tarsi, phalangeal formulae and digit length and proportions. To facilitate visual inspection of these elements, all hindlimbs are drawn to a common tibial length, except for the stem diapsid *Petrolacosaurus*, in which the epipodials are greatly elongate.

In terms of pes size relative to the tibia, the East Kirkton taxa *Balanerpeton*, *Eucritta*, and *Silvanerpeton* (Fig. 4a, b, d) are similarly proportioned. In contrast, *Eldeceeon* and *Westlothiana* (Fig. 4c, e) exhibit somewhat larger pedes. *Kirktonecta* has proportionally the largest pedes of all (Fig. 4f). *Termonerpeton* (Fig. 4g) has a pes of similar size to the first three taxa except that digit IV is relatively much longer than in any of the others, with an exceptionally large metatarsal IV. In all those taxa in which digit IV is fully preserved, it is the longest, especially in *Eldeceeon* and *Kirktonecta*, but in none does it approach in size and proportions that of *Termonerpeton*. The illustrated limbs also differ from one another in the degree of ossification of the tarsal bones. Most taxa except *Eucritta* have some indication of ossified tarsal elements, and some of them, including *Balanerpeton* and *Silvanerpeton*, show a complete or almost complete tarsal set. *Kirktonecta* does have an ossified tarsus, but specimen preservation does not allow us to identify individual elements. The phalangeal count, where known, also varies: 22343 in *Balanerpeton*[2]; 223?? in *Eucritta*[12]; 23455 in *Silvanerpeton*[4]; 23454 in *Eldeceeon*[6], *Kirktonecta*[15], *Termonerpeton*, and *Westlothiana*[7].

In addition, we compared the pedes of East Kirkton tetrapods with those of seven other taxa (Fig. 4h–n): one earlier, *Pederpes*[24]; one almost contemporary, *Caerorhachis*[25]; four later Carboniferous, *Greererpeton*[26], *Hylonomus*[27], *Tuditanus*[22], and *Petrolacosaurus*[28]; and one early Permian, *Archeria*[29]. Of these, *Greererpeton* has relatively the smallest pes. In most, digit IV is the longest, though in *Pederpes* and *Caerorhachis* it is incomplete. The pes of *Caerorhachis* was originally restored with only three phalanges in digit IV[30]. This is probably incorrect and would be unusual in Carboniferous tetrapods. The pes of the anthracosaur *Archeria* was originally reconstructed with digit V as the longest[29], but again this is unusual among later Carboniferous and early Permian tetrapods and we suspect that digits IV and V have been transposed, and Romer himself expressed doubt about this reconstruction[29]. In either case, the phalangeal formula of *Archeria* is similar to that of the East Kirkton anthracosaur *Silvanerpeton*, as 23455.

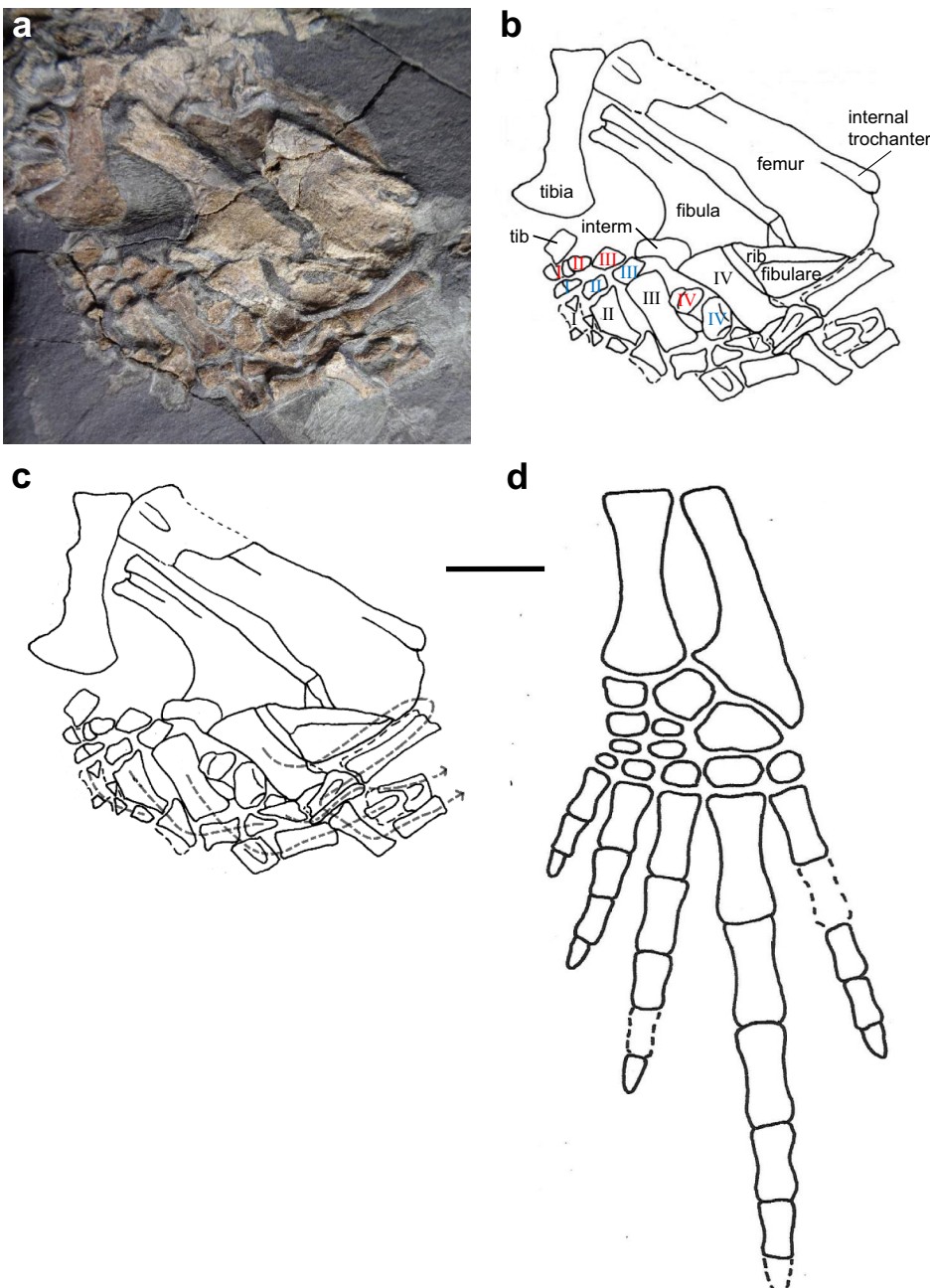

**Fig. 2 *Termonerpeton makrydactylus* gen. et sp. nov. left hindlimb of UMZC 2019.1. a** Specimen photograph, showing close-up view of hindlimb skeleton, **b** Interpretive drawing, with centralia, distal tarsals, and metatarsals indicated by red, blue, and black Roman numerals, respectively, **c** Interpretive drawing with dashed lines connecting elements of individual digits, **d** Reconstruction of left tibia, fibula and pes. Scale bars 10 mm. Abbreviations: interm intermedium, tib tibialia.

Among Carboniferous tetrapods, temnospondyls such as *Balanerpeton* and colosteids such as *Greererpeton* show a digit IV that is somewhat longer than the others, but metatarsal IV is very similar in length and breadth to the adjacent metatarsals. In anthracosaurs, digit IV is the longest, but again metatarsal IV is not significantly broader than adjacent metatarsals. This is also the case in the early amniote *Hylonomus* and the microsaur *Tuditanus*. Among the taxa illustrated here, *Termonerpeton* shows a strikingly similar pes to that of the Late Pennsylvanian araeoscelidian diapsid *Petrolacosaurus* (Fig. 4n). In both, metatarsal IV is significantly longer and stouter than others and forms part of a similarly long digit IV. In early amniotes, an elongate digit IV coupled with an elongate metatarsal IV is a common occurrence in other taxa, such as prothothyridids (e.g. *Anthracodromeus*[31]), basal araeoscelidians (e.g. *Spinoaequalis*[32]), younginids (e.g. *Youngina*[33]), saurians[33], and basal synapsids (e.g. *Heleosaurus*[34–36]), among others.

Based upon available evidence, an elongate digit IV is likely to be the plesiomorphic condition for crown amniotes, being present in *Hylonomus*, *Paleothyris*, and *Petrolacosaurus* (Fig. 4l, n), and shortening of this digit certainly represents a derived feature. In later crown amniotes, the conditions vary, with larger, heavier-bodied tetrapods such as dicynodonts and diadectids having generally shorter toes and adopting a more clearly plantigrade posture. An elongate metatarsal IV and associated digit, however, are not universal among Palaeozoic amniotes, and modifications of

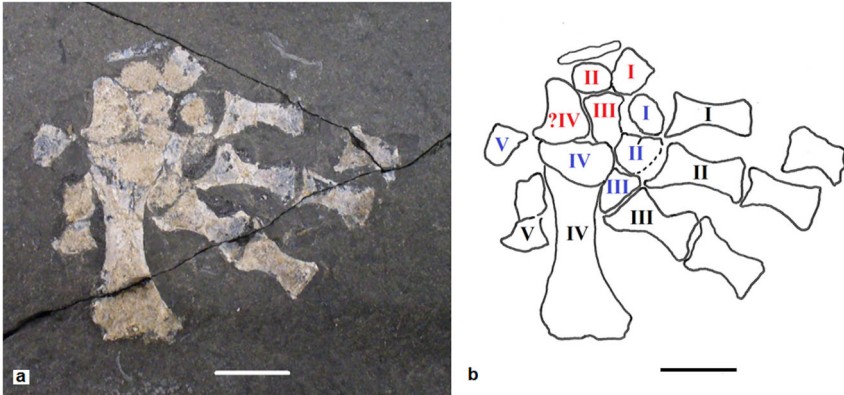

**Fig. 3 *Termonerpeton makrydactylus* gen. et sp. nov partial pes, attributed specimen NMS G.1992.22.1. a** Specimen photograph, **b** with centralia, distal tarsals, and metatarsals indicated by red, blue, and black Roman numerals, respectively. Scale bars 10 mm.

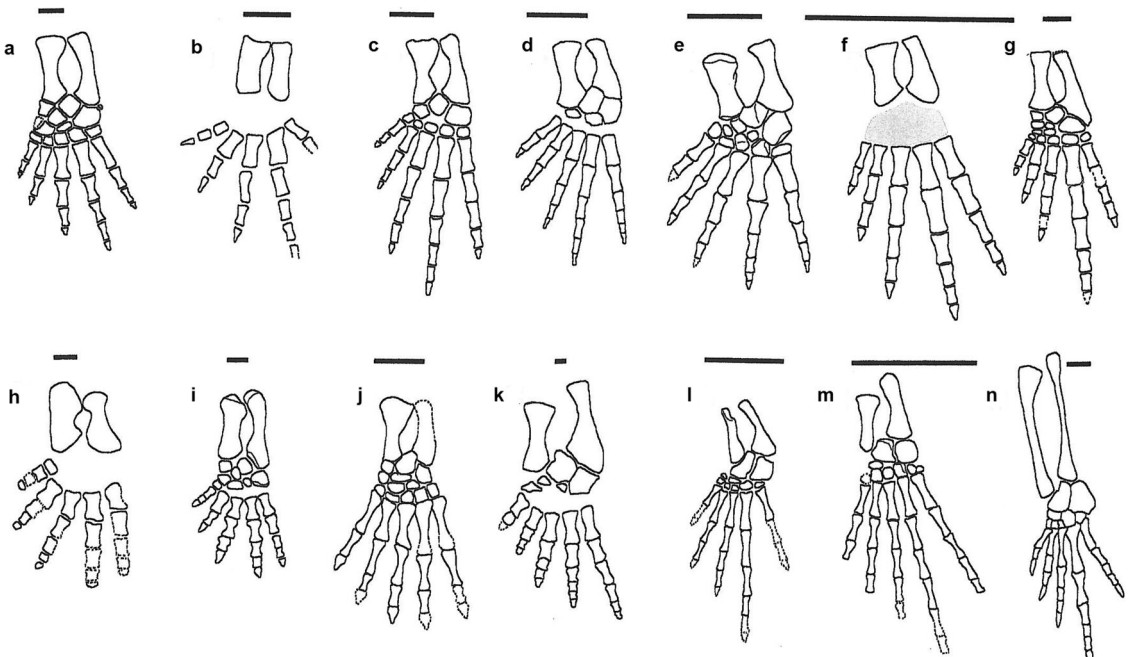

**Fig. 4 Comparison of the left tibia, fibula, tarsus, and digits of early tetrapods. a** *Balanerpeton* after 2, **b** *Eucritta* after 12, **c** *Eldeceeon* after 6, **d** *Silvanerpeton* after 4, **e** *Westlothiana* after 7, **f** *Kirktonecta* original, see 15 (the grey area marks the estimated position and extent of the tarsus), **g** *Termonerpeton*, **h** *Pederpes* after 24, **i** *Greererpeton* after 27, **j** *Caerorhachis* after 31, **k** *Archeria* after 30, **l** *Hylonomus* after 28, **m** *Tuditanus* after 22, **n** *Petrolacosaurus* after 29. Drawn to the same tibial length apart from **n**. Scale bars 10 mm.

these conditions occur repeatedly across clades. For instance, in the eureptile captorhinid *Eocaptorhinus*, digit IV is also the longest, but the length of metatarsal IV does not greatly exceed that of other metatarsals[37]. The same is true of some early Permian clades, including seymouriamorphs (e.g. *Seymouria*[38]; *Discosauriscus*[39]), and diadectids (e.g. *Diadectes*[40]), although in the diadectomorph *Orobates* digit III is a little longer than digit IV[41]. Among synapsids, dicynodonts such as *Diictodon*[42] and caseids[43], to name a few, have five pedal digits of approximately uniform length.

We further point out that, while digit IV attains a certain degree of elongation in other early tetrapod groups, such as temnospondyls, in none of them do the relative proportions of this digit (where known) compare to those of several stem and crown amniotes (Fig. 4).

**Phylogenetic relationships**. The results of various phylogenetic analyses lend some support to the interpretation of

*Termonerpeton* as a stem amniote, despite its uncertain placement in the unweighted character parsimony analysis (Fig. 5a). In the latter analysis, *Termonerpeton* appears in a polytomous node alongside baphetids (*Eucritta*; *Baphetes*; *Megalocephalus*), temnospondyls (*Balanerpeton*; *Dendrysekos*), the anthracosauroids *Eldeceeon* and *Silvanerpeton*, and the problematic *Caerorhachis*. In all other analyses—implied weights, reweighted characters, and Bayesian—*Termonerpeton* is placed on the amniote stem group, albeit in different positions, among a diverse array of 'reptiliomorph' clades and grades. In the implied weights analysis (Fig. 5b), *Termonerpeton*, *Silvanerpeton*, and *Eldeceeon* form a monophyletic group branching crownward of chroniosaurs plus anthracosaurs and anti-crownward of paraphyletic gephyrostegids. In the reweighted analysis (Fig. 5c), *Termonerpeton* and *Caerorhachis* appear as successive sister taxa, in that order, to monophyletic anthracosaurs. In the Bayesian analysis (Fig. 5d), the amniote total group receives moderate support with a credibility value (c.v.) of 76 with *Caerorhachis* as the most

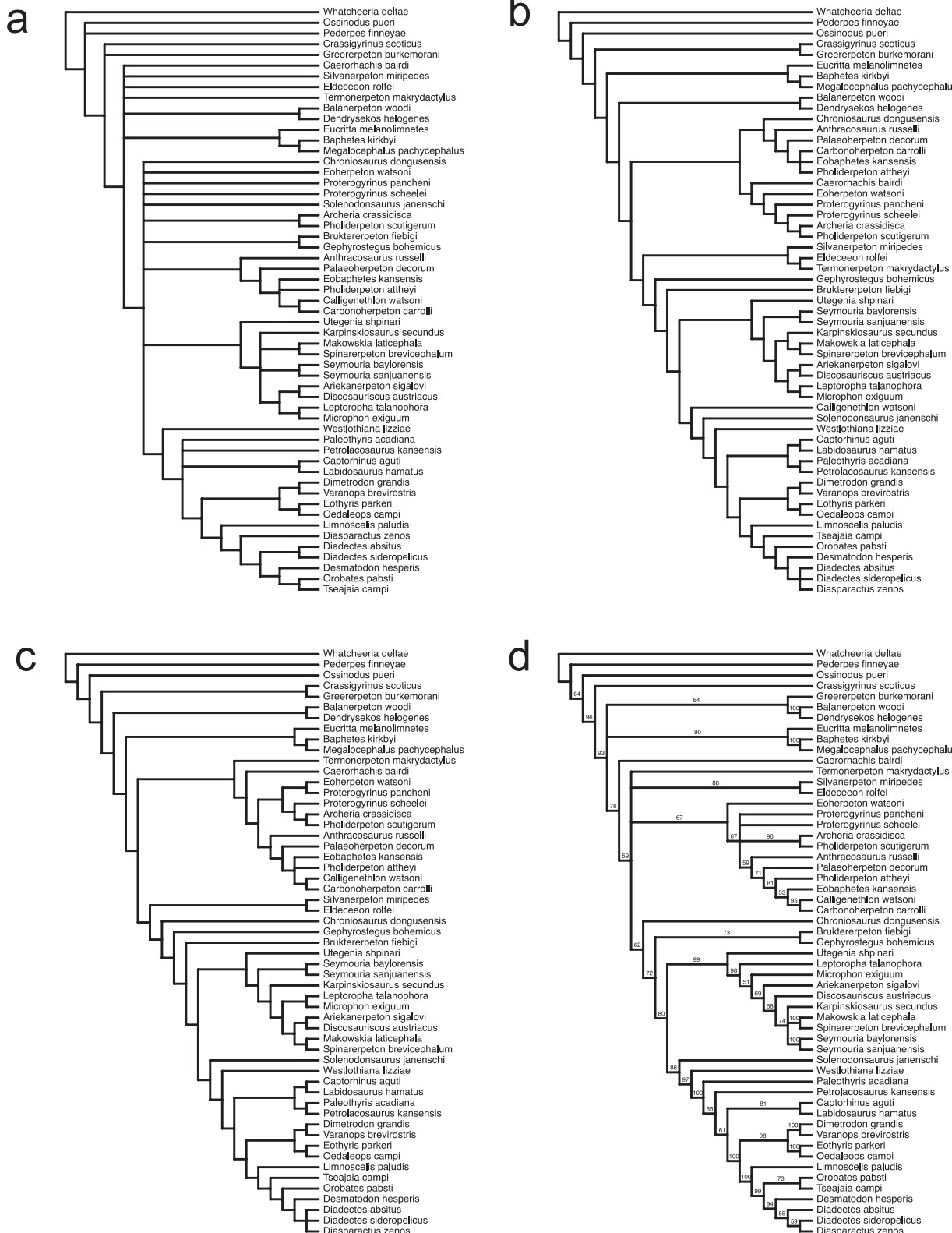

**Fig. 5 Results of phylogenetic analyses. a** Strict consensus of 120 shortest trees from unweighted analysis (tree length = 1286 steps, ensemble consistency index C.I. = 0.2738 without uninformative characters, ensemble retention index R.I. = 0.5768), **b** Single tree from implied weights analysis (tree length = 1298 steps, Goloboff fit = −202.59266, C.I. = 0.2712, R.I. = 0.5713), **c** Single tree from reweighted analysis (tree length = 212,68965 steps, C.I. = 0.4755, R.I. = 0.774), **d** Bayesian topology with branches reporting credibility values.

plesiomorphic stem amniote. Crownward of *Caerorhachis* is a polytomy with low support (c.v. = 59) that subtends *Termonerpeton*, a clade consisting of *Eldeceeon* plus *Silvanerpeton*, a clade of anthracosaurs, and a clade that includes all remaining taxa. In crownward succession, these taxa include chroniosaurs, gephyrostegids, seymouriamorphs, *Solenodonsaurus*, and *Westlothiana* as successive sister groups to a strongly supported (c.v. = 100) clade containing diadectomorphs, synapsids, and eureptiles. Although eureptile monophyly is not retrieved, strong support (c.v. = 100) is given to the branch subtending diadectomorphs plus synapsids[44].

## Discussion

**Morphofunctional and ecological considerations**. The East Kirkton biota includes a diverse tetrapod fauna with unique morphofunctional and ecological specializations. In particular, the diversity of their limb and digit proportions and phalangeal constructions suggests a variety of locomotory modes. *Kirktonecta* features aquatic adaptations[15], as evidenced by its deep tail and paddle-like hindlimbs. Five taxa (*Balanerpeton*; *Eucritta*; *Eldeceeon*; *Silvanerpeton*; *Westlothiana*) appear to have been more fully terrestrial or semi-terrestrial[2–7,12–14], as suggested by the presence of fully ossified and tightly appressed tarsal bones (such as are observed in *Balanerpeton*, *Silvanerpeton* and *Westlothiana*), robust and strongly ossified limbs with well-developed muscle attachment processes, finished margins of dermal pectoral elements, and absence of lateral line canals[3]. The limbless *Ophiderpeton* is also considered to be terrestrial or semi-terrestrial, due to the lack of sensory canals[14]. Except for *Ophiderpeton*, all other East Kirkton tetrapods exhibit a five-digit pes, a condition first recorded in the Tournaisian[24]. There is no evidence of polydactyly in any other known Carboniferous tetrapod, unlike in late Devonian taxa (e.g. *Acanthostega*; *Ichthyostega*; *Tulerpeton*)[45].

Although *Termonerpeton* is known from incomplete material, its hindlimb is known in sufficient detail and permits reasonable inferences about its locomotion. Comparisons with modern analogues from among squamate reptiles—a focal group in biomechanical studies of amniote locomotion—are helpful in this respect. In several extant lizards, a large pes with a long metatarsal IV and associated digit occur commonly and are often highly specialized. For example, in runners they act, *inter alia*, as an energy-saving mechanism generally linked to increased stride length[46]. Digit IV is the last to leave the ground during a stride cycle and confers extra leverage when the animal moves fast. In addition, the tendons of elongate digits are known to operate as springs[47], further assisting energy-saving during running. Ruta et al.[6] proposed a similar mechanism in the East Kirkton anthracosauroid *Eldeceeon* (Fig. 4c). The unusually large pes of this tetrapod was hypothesized to increase stride length and to facilitate forward propulsion at the end of each stride cycle when the pes was fully extended backwards. In *Eldeceeon*, however, digit IV is comparatively less long and robust—relative to digits III and V–than its homologue in *Termonerpeton*. If indeed *Termonerpeton* used its pes in a similar way, then its digit IV is likely to have functioned as the principal, or sole, lever during locomotion.

It is noteworthy that a long digit IV also provides remarkable biomechanical versatility in extant squamates. As an example, it allows the Zebra-tailed lizard to run on compact and granular substrates[46]. The varied terrain of the East Kirkton environment[17–21] suggests that, perhaps, the pes of *Termonerpeton* afforded this animal some degree of biomechanical versatility. However, it is possible to rule out cursorial habits in *Termonerpeton*. Its hindlimb lacks the suite of traits usually associated with running capabilities in several extant and fossil tetrapods, such as the basilisk lizard and the Early

Permian bolosaurid parareptile *Eudibamus*, to name two well-known examples. The traits in question include, *inter alia*, very elongate, subparallel, and slender tibia and fibula delimiting a narrow interepipodial space, rod-like and closely packed metatarsals III-V, and elongate and gracile proximal phalanges of digits III–V. The construction of the *Termonerpeton* pes, particularly its sturdy and wide tarsus and the extensive sutural contact between its fibulare and the broad distal extremity of its fibula, appears to us to be indicative of a graviportal stance.

Indirect support for this conclusion comes from a consideration of the estimated body size of this animal. If we assume that *Termonerpeton* scaled isometrically with *Eldeceeon* and that both taxa had comparable presacral:femur length ratios, then *Termonerpeton* is estimated to have reached a presacral length of ~75 cm, more than twice the length of the *Eldeceeon* holotype[5] (~35 cm; 18 mm femur length). If we use *Balanerpeton* as a reference taxon, we obtain a more modest figure. One of the most complete paratypes of *Balanerpeton* measures 17.5 cm in length, with a femur of 14.23 mm[2]. Scaling *Termonerpeton* to this specimen gives an estimated length of ~48 cm. Based upon these figures, we conclude that *Termonerpeton* was probably much more heavily built than other East Kirkton tetrapods. However, it remains a possibility, that the elongate digit IV enabled *Termonerpeton* to move more quickly than its heavy build might suggest.

**Evolutionary implications**. The major finding of our study is that *Termonerpeton* is the earliest known tetrapod to exhibit a fundamentally amniote-like pes[48], typified by elongate metatarsal IV and corresponding digit IV, and by enlarged intermedium and fibulare that occupy more than half of the proximal moiety of the tarsus. More proximal segments of the hindlimb appear less specialized. Thus, the tibia and fibula are relatively stout, and the robust femur has a weakly pronounced mid-shaft waisting. This combination of plesiomorphic and apomorphic traits suggests that the emergence of fully terrestrially adapted limbs among stem amniotes followed a less regular pattern than previously surmised. Both embolomeres and gephyrostegids retain unspecialized tibia and fibula which, discounting minor proportional differences, resemble their homologues in *Termonerpeton*. In contrast, their femora appear more derived than those of *Termonerpeton* in that they display a more pronounced waisting of the shaft. Their pedes, where known, demonstrate a gradual acquisition of amniote-like characteristics[48]. These include enlargement of the proximal part of the tarsus, progressive elongation of digits (especially digits III–V), and the transverse expansion of the laterodistal extremity of the fibula. These features also occur in seymouriamorphs[38,39], diadectomorphs (albeit modified towards metatarsal and phalanx shortening)[40,41], and several basal crown amniotes[48]. Some of these features have recently been recognised in fossil footprints from the late Carboniferous and Permian[49,50]. However, given the range of hindlimb morphologies present at East Kirkton, and the amniote-like pes found in *Termonerpeton*, we would advise caution when ascribing particular footprints to specific clades.

## Methods

**Preparation and visualization**. The holotype specimen was prepared using an Emax Evolution Grinder and Polisher fitted with a chuck to accommodate a tungsten carbide rod. Photography by JAC and TRS used a Panasonic Lumix DMC-LZ5. Specimen drawings by JAC and TRS were executed using a camera lucida and followed by image processing with Photoshop CC 2017 and 2019.

**Nomenclatural acts**. This published work and the nomenclatural acts it contains have been registered in ZooBank, the proposed online registration system for the International Code of Zoological Nomenclature (ICZN). The ZooBank LSIDs (Life Science Identifiers) can be resolved and the associated information viewed through

any standard web browser by appending the LSID to the prefix 'http://zoobank.org/' The LSID for this publication is: 72B609CB-DCC1-4BDA-9494-BB97BBEB70E4.

**Phylogenetic analyses**. To evaluate the phylogenetic position of *Termonerpeton*, we coded this taxon using a slightly expanded version of a recently published taxon-character data matrix[44] which was subjected to maximum parsimony and Bayesian inference analyses by MR. Prior to parsimony analyses, the data matrix was inspected for possible occurrences of 'rogue' species[51] causing loss of resolution among other, more stable species. No 'rogue' taxa were detected using the 'safe_taxonomic_reduction' function of the package Claddis[52] in the R (v. 4.1.1) environment for statistical computing (https://cran.r-project.org). In the case of parsimony analyses, we explored the results from unweighted, posteriorly weighted and standard implied weighted characters[53]. Posterior weighting employed the maximum value (best fit) of the rescaled consistency index of each character, such as was obtained from the original unweighted analysis, whereas implied weighting used a value of 6 for the constant of concavity K[53]. All parsimony analyses were performed in PAUP* v. 4.0a build 169 (https://paup.phylosolutions.com)[54] using identical settings, as follows: heuristic search method; tree bisection-reconnection branch-swapping algorithm with 10,000 random stepwise taxon addition sequences, holding a single tree in memory at each replicate; 10 consecutive rounds of branch-swapping applied to all trees stored in memory from this initial search, but with the option for saving multiple trees in effect. Node support was evaluated with bootstrap[55] and jackknife[56], in each case with 10,000 random character resampling replicates using the fast stepwise addition option.

The Bayesian inference analysis was carried out with MrBayes v. 3.2.6 (http://nbisweden.github.io/MrBayes/index.html)[57] using the standard data type option for morphological characters, and an equal-rate model of character-state change. The analysis employed four chains with $10^7$ generations, sampling every 1000 generations, discarding 25% of the obtained samples, and storing branch lengths alongside clade credibility values. Convergence was regarded as satisfactory, with the values of the Potential Scale Reduction Factor[58] approaching 1.

**Reporting summary**. Further information on research design is available in the Nature Research Reporting Summary linked to this article.

## Data availability

The holotype of *Termonerpeton makrydactylus* is accessioned in the University Museum of Zoology Cambridge (UMZC) register number 2019.1. The attributed specimen is accessioned in the National Museums of Scotland (NMS) register number G.1992.22.1.

The data matrix used in this study is available in Supplementary Data 1 and the character list in Supplementary Note 1 with the Supplementary Information file.

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

## Acknowledgements

We thank Daniel Delbarre who prepared the UMZC specimen during an undergraduate summer project and Sarah Wallace-Johnson (formerly Finney; Earth Sciences Department, University of Cambridge), for the loan of the Emmax Evolution Grinder and Polisher. We are grateful to Matt Lowe (University Museum of Zoology Cambridge) for access to UMZC 2019.1 and Nick Fraser and Stig Walsh (National Museums Scotland) for the loan of NMS G 1992.22.1. We thank our journal editor, Luke Grinham, two anonymous reviewers and Torsten Scheyer (University of Zurich) for their constructive criticism and helpful remarks. Tim Smithson thanks Jason Head and Rebecca Kilner (University Museum of Zoology Cambridge) for access to research facilities. This project was supported by a John Templeton Foundation Grant (No 61408) awarded to M.R.

## Author contributions

J.A.C. conceived the study. J.A.C. and T.R.S. photographed the specimens and prepared the figures. M.R. undertook the phylogenetic analyses. All authors described and analysed the fossil material, interpreted the data, and wrote the manuscript.

## Competing interests

The authors declare no competing interests.
