## [Peer Review File · Communications Biology]

Reviewers' comments:

Reviewer #1 (Remarks to the Author):

This manuscript described a new genus of early tetrapods from the late Mississippian East Kirton Limestone, increasing the taxonomic diversity and morphological disparity of a well-known Carboniferous terrestrial fauna. The referred fossils are well described and illustrated, and the discussions and comparisons with other related taxa are convincing. As such, I recommend its publication in *Communications Biology*. However, I have following minor comments for consideration.

- 1) Being the earliest known tetrapod to exhibit a fundamentally amniote-like pes construction, the new form, however, cannot be definitely referred to as a stem-amniote, as shown by the parsimony analysis. This point has not been clearly stated in the abstract.
- 2) Differential diagnosis might be better than the present diagnosis. Which characters separate the new taxon from other related tetrapods?
- 3) The reference citations in the main-text and figures need careful edition.

Reviewer #2 (Remarks to the Author):

The manuscript describes a new tetrapod from the Early Carboniferous period and uses it as the basis for a new analysis of the early evolution of the tetrapod pes. This is an important discovery because of the relative lack of tetrapod fossil material from this time period, when tetrapods first became terrestrially capable and the amniote and amphibian lineages diverged. The pes of *Termonerpeton* is notable for being the earliest example of "amniote-like construction," mainly due to an elongate 4th digit.

The manuscript is well written with high quality figures. It describes an important discovery that bears on the evolution of terrestrial locomotion in tetrapods. The impact might be enhanced by a more focused discussion (specific comments below).

Specific comments

In the results section it is sometimes unclear whether the dorsal or ventral aspect of a bone is being described.

Line 98 refers to a probable acetabulum; please label in Figure 1b and describe if possible.

Line 125: "the proximal end [of the femur] also appears concave" i.e., the ventral aspect of the femoral head? The proximal end looks convex in the photo and illustration. Please clarify.

The discussion section focuses mainly on the pes. In general, I would appreciate more explicit discussion of the relevance of these characters to the issues brought up in the introduction, namely early appearance of certain hindlimb specializations, underappreciated diversity among the tetrapods of the East Kirkton locality, and terrestrial adaptations.

Lines 212-213: "experimenting ... probably in response to the varied terrain of the terrestrial environment" - because it was new to them or because terrestrial environments are intrinsically

more varied than aquatic ones? Or is the East Kirkton environment particularly variable?

What was the purpose of comparing the pedal proportions with that of other Carboniferous tetrapods? This section seems unfocused. I suggest re-organizing it around specific predictions related to ecology, phylogeny, etc.

Lines 253-259: I am happy to see a consideration of locomotor function of D4. However, this paragraph is vaguely worded and the point is unclear. If an elongate MT4 and D4 are "nearly universal" in modern lizards, how can they be associated with different functional characteristics? Presumably it has to do with extreme elongation, but how is that defined? Why is varied substrate the preferred functional hypothesis for its presence in Termonerpeton - because it doesn't aid in stride length or leverage during slow locomotion?

Lines 273-274: "fundamentally amniote-like pes construction" meaning 5 toes and elongate MT4/D4?

I suggest a few sentences at the end of the discussion summing up the big-picture implications of the discovery. For example, to me the morphology of this fossil seems to suggest that adaptations for more versatile/advanced terrestrial locomotion were present at the very base of the amniote lineage. I'd love to hear what the authors feel is the most important takeaway or most interesting possibility raised by this exciting discovery.

Reviewer #3 (Remarks to the Author):

I generally encourage publication of the described specimens because of our still limited understanding of early tetrapod evolution, but I have some major points of criticism that need to be addressed.

- 1) the quality and labeling of the figures, especially of Fig. 1 and 2 need to be improved
- 2) all elements of the tarsus need to be clearly identified and labelled because much of the discussion is centered around these.
- 3) In my opinion, the interpretation of much of the pes is rather speculative, and other bones are damaged or crushed, thus providing limited information. The authors should reassess whether the material is sufficient for describing a new species and genus.
- 4) Additional images (with angled/side light) or coating with ammonium chloride should be considered - not mandatory though.

I have made other comments directly on the PDF.

Torsten Scheyer

A new Mississippian tetrapod showing early diversification of the hind limbs

Jennifer A. Clack¹⁺, Timothy R. Smithson^{1*} & Marcello Ruta²

Author affiliations

1 University Museum of Zoology Cambridge, Downing Street, Cambridge, CB2 3EJ, UK.

2 School of Life Sciences, University of Lincoln, Joseph Banks Laboratories, Green Lane,

Lincoln, LN6 7DL, UK.

+ Deceased

*Corresponding author ts556@cam.ac.uk

Summary

The terrestrial tetrapod fauna from the late Mississippian East Kirkton Limestone consists of

a taxonomically diverse assemblage that includes the earliest known members of stem

Amphibia and stem Amniota. Here we name and describe a new stem amniote from East

Kirkton with an unusual hind limb morphology, *Termonerpeton makrydactylus*. We

compared its phalangeal formula, digit length and proportions with those of six other

tetrapods from the East Kirkton Limestone and seven other Palaeozoic species.

*Termonerpeton* shares with many of the earliest amniotes a 5-digit pes with an elongate digit

IV. The morphology of its pes is most similar to that of the Late Pennsylvanian eureptile

*Petrolacosaurus* in having a long and stout metatarsal IV followed by a similarly long digit

IV. The East Kirkton fauna shows a remarkable range of hind limb morphologies suggesting

that the earliest known terrestrial tetrapods were experimenting with limb proportions

adapted to a wide variety of terrestrial substrates. Bayesian and weighted parsimony analyses

place the new taxon in the amniote total group, albeit in different positions, among the

earliest diverging stem amniote clades.

The Carboniferous tetrapod fauna from the East Kirkton Limestone of the Bathgate Hills in
Scotland provides a unique window onto the diversity of terrestrial vertebrates at the end of
the Mississippian, about 336 mya. The seven East Kirkton tetrapods described and named so
far are both taxonomically diverse and morphologically disparate, exhibiting different body
shapes, vertebral constructions, and limb proportions. This level of diversity among tetrapods
is not encountered again until the mid-Pennsylvanian some 25 myr later, by which time
several tetrapod clades, in particular crown Amniota, are well established and diversified
(1Mann *et al.* 2020). Among the tetrapods represented at East Kirkton are the temnospondyl
*Balanerpeton* (2Milner & Sequeira 1994), the anthracosaurs *Silvanerpeton* (3Clack 1994,
4Ruta & Clack 2006) and *Eldeceon* (5Smithson 1994, 6Ruta *et al.* 2020), and the amniote-
like *Westlothiana* (7Smithson *et al.* 1994). In some recent studies, (8, 9Clack *et al.* 2016,
2019, 10Pardo *et al.* 2017), these taxa have been placed phylogenetically as the earliest
known stem Amphibia (*Balanerpeton*) and stem Amniota (*Westlothiana*; *Silvanerpeton*;
*Eldeceon*; but see 11Marjanović & Laurin 2019), and thus provide a minimum age estimate
for the origin of crown Tetrapoda (12Clack 1998, 8Clack *et al.* 2016). Other East Kirkton
tetrapods include the baphetid *Eucritta* (12Clack 1998, 132001), the aïstopod *Ophiderpeton*
(14Milner 1994), and the probable microsaïr *Kirktonecta* (15Clack 2011). Except for
*Ophiderpeton*, all these taxa are among the most plesiomorphic members of tetrapod clades
that are more commonly found throughout the later Palaeozoic.

In this paper, we name and describe a new tetrapod from East Kirkton, based upon
postcranial remains that show unusual specializations to the pelvic and hind limb skeletons,
until now observed only in much later tetrapods. This new taxon prompts a reconsideration of
the significance of the East Kirkton fauna for our understanding of the radiation of tetrapods

during the Viséan stage of the early Carboniferous and, more broadly, the early evolution of
 tetrapod terrestrial adaptations, particularly in their appendicular skeleton.

Results

Systematic Palaeontology

Tetrapoda Jaekel, 1909 (*vide* 16Sues 2019)

Family undesignated

*Termonerpeton makrydactylus* gen. et sp. nov. (Fig. 1)

**Etymology.** *Genus*: from τέρμων (térmon) meaning boundary and ερπετό (erpetó) meaning
 ‘crawler’, referring to the field boundary walls near the East Kirkton quarry where the late
 Stan Wood initially discovered fossils from the East Kirkton Limestone and from where the
 type specimen may have been collected; *species*: from μακρός (makrós) meaning ‘elongate’
 and δάχτυλο (dáchtylo; more precisely, δάχτυλο ποδιού, dáchtylo podiou) meaning ‘toe’,
 referring to the very long pedal digit IV.

**Holotype.** University of Cambridge Museum of Zoology (UMZC) 2019.1. A partial tetrapod
 postcranium, preserving both pelves, a femur, fibula, tibia, and an almost complete but
 **disrupted** pes. Closely associated with the appendicular elements are dorsally open hoop-
 shaped centra, a few neural arches, ribs, and a section of articulated gastralia.

**Locality and horizon.** East Kirkton quarry, near Bathgate, Scotland, UK. East Kirkton
 Limestone, Bathgate Hills Volcanic Formation. Exact horizon unknown. Brigantian, Viséan,
 early Carboniferous (=Mississippian) (17Smithson 1985).

**LSID number.** To be added

**Diagnosis.** Ilium with drawn out, flat, blade-like dorsal process and long post-iliac process.
 Short puboischiadic plate with almost vertical anterior margin. Stout femur longer than
 puboischiadic plate. Large interepipodial space between tibia and fibula. Well ossified tarsus

comprising tibiale, fibulare, intermedium, four centralia, and five distal tarsals. Very large,
stout, and elongate metatarsal IV. Robust and long pedal digit IV. Pedal phalangeal formula
23454.

**Attributed specimen.** National Museums Scotland (NMS) 1992.22.1. An articulated,
partially complete tetrapod pes. Unit 82, East Kirkton Limestone, East Kirkton quarry, near
Bathgate, Scotland, UK.

**Remarks.** The holotype was collected by Stan Wood and donated to the UMZC, probably in
the 1990's, although details of its collection were, unusually, not recorded by him. Possible
reasons for this are that the specimen derived either from one of the spoil heaps from the old
quarry workings of the locality or from one of the boundary walls that Stan had bought and
collected from before the main quarry was discovered (18Smithson & Rolfe 2018). However,
the matrix lithology suggests the specimen may have originated from Unit 82 (19Rolfe *et al.*
1994a). The unusual circumstances of fossil preservation, in a locality affected by volcanic
activity and including a mineral-rich lake fed by warm or hot springs, have been described in
a series of papers in Rolfe *et al.* (201994b) and summarized in Clack (212017). As well as the
earliest known terrestrial tetrapods, the East Kirkton quarry has also yielded fish, arthropods,
and plants (20).

Specimen description

**Appendicular skeleton.** Both pelves are preserved, one mainly as a natural mould. Both
puboichiadic plates are short and deep, with almost vertical anterior margins to the pubis (Fig
1). In one, the surface of the puboischiadic plate is strongly convex, in the other it is strongly
concave. The concave plate may belong to the left pelvis, with the concavity indicating the
acetabulum. Both iliac processes of the presumed right ilium are overlain by a neural arch
and part of the femur and cannot be seen. The presumed left ilium shows a long, posteriorly

pointing post-iliac process that extends as far backward as the posterior edge of the ischium.
It retains the stump of a dorsal iliac process, continued in natural mould as a mediolaterally
flattened blade-like structure. Both processes sit above a short neck. The dorsal iliac process
is proportionally longer than in other tetrapods and its knife-like shape appears to be unique.
The angle between the two processes is much more acute than in most other tetrapods, and
the nearest comparison is with the divided iliac process of the microsauro *Ricnodon* (Carroll
& Gaskill 1978) which, however, could merely represent a bifid post-iliac process.

Two gaps in ossification are taken as evidence of an ilio-ischiadic suture half-way
down the posterior margin on the left pelvis and an ilio-pubic suture half-way down the
anterior margin of the right pelvis (Fig. 1). There is no evidence of a puboischiadic suture,
although a shallow depression along the ventral margin of the left puboischiadic plate
probably marks the junction between pubis and ischium. The complete left puboischiadic
plate is 20 mm deep behind the ilium and 30 mm long, with the pubis contributing about one
third of its length and the ischium the remaining two thirds. The anterior margin of the pubis
is almost vertical. The dorsal margin of the ischium is shallowly convex for half its length
before extending posteroventrally to meet the upturned posterior extremity of the ischium's
ventral margin. There is no evidence as to the angle at which the two pelvic plates met at the
symphysis, which would affect the position of the acetabulum relative to the substrate, and
thus the effective resting posture of the hind limb.

Figure 1 here

The left femur is at least 39 mm in length, and longer than the puboischiadic plate.
The entire bone is crushed and its distal end lies partly beneath one of the pelvic halves and a
neural arch, so that its features cannot easily be made out. A possible intercondylar groove

may be present distally, and the proximal end also appears concave. The femur itself is robust
with little waisting at mid-shaft. A small internal trochanter lies near its proximal end.

The left fibula is approximately 26 mm long along its lateral margin. Its proximal end
is narrow and grooved. Its broad and strongly flared distal end suggests a broad contact with
the tarsus. The medial turn of the distal end indicates a large interepipodial space. The left
tibia is about 20 mm long, slender, and shallowly waisted at mid-shaft. It is not clear which
end is proximal and which distal, although probably the proximal is the broader. The tibia is
probably more than half the length of the femur. Based upon the femur and tibia lengths, and
omitting the ankle, the above figures indicate a total limb length of about 65 mm, assuming a
fully extended limb.

Most of the morphology of the left pes is preserved, showing many well-ossified
tarsal bones (Fig. 2). Several of these, including possible distal tarsals II and III lie more or
less in anatomical continuity relative to metatarsals II and III, respectively. Other tarsal
elements, including possible fibulare, tibiale, centrales, and distal tarsals, are illustrated in
Figure 2. Metatarsal IV lies in anatomical position relative to metatarsals II and III, and is
significantly larger than the latter, measuring 7 mm in length. The presumed first phalanx of
pedal digit IV lies at an angle of nearly 90 degrees to metatarsal IV. It is long and slender,
indicating an unusually elongate fourth pedal digit. Together, the pedal elements suggest a
relatively large foot.

Figure 2 here

An array of about 12 phalanges is preserved. They are all disrupted and, like the first phalanx
of pedal digit IV, also mainly lie at right angles to metatarsals III and IV. An acutely angled
pointed unguis, possibly associated with digit II, is also visible. A further two phalanges have

been displaced and rest along the anterior edge of the left pelvis. The preservation of the pes
suggests it was strongly flexed either at death or from tissue shrinkage thereafter. An isolated
metatarsal, presumably from the other, missing foot lies at a distance near the edge of the
block.

A second pes specimen, NMS G.1992.22.1 (Fig. 3), may belong to *Termonerpeton*,
although it is from a larger individual. It shows five metatarsals of which the fourth is much
longer than the other four, with metatarsal V being the smallest. There are three phalanges,
plus five distal tarsals. A D-shaped element closely associated with three centrales could be
either a fibulare, a displaced intermedium, or centrale IV.

Figure 3 here

**Axial skeleton.** Where visible, neural arches have short neural spines and prominent
zygapophyses, but their shape is hard to assess as none is well preserved. The element
overlying part of the right pelvis and the femur is 7 mm high in total. Numerous dorsally
open, hoop-shaped centra about 5 mm in diameter are visible, as well as a few small, oval,
shallowly curved elements (Fig. 1). Without further evidence it is uncertain which of these
elements are intercentra and which pleurocentra, though we assume that the larger elements
are pleurocentra.

The preserved ribs are slender and curved, and include trunk ribs, a possible presacral
rib, a possible sacral rib, and a possible postsacral rib. This is long but more or less straight.

A bone situated among a cluster of centra, somewhat distant from the other tarsal bones, was
originally interpreted by us as a possible fibulare, similar to the fibulare in *Proterogyrinus*
(23Holmes 1984). However, it might also be interpreted as a sacral rib. If so, its morphology
is unique. It is short and widens distally into a fan-shaped structure but does not appear to
have a bifid proximal end, unlike the sacral rib in *Proterogyrinus* (23Holmes 1984).

Three haemal arches are present, one still attached to its half-hoop centrum, a second
slightly longer, and a third very short and presumably from a more posterior region of the tail.

Discussion

**Hind limb morphology.** The exceptional preservation of tetrapods in the East Kirkton
Limestone provides a unique opportunity to study portions of the skeletal anatomy that are
otherwise poorly preserved or absent among Mississippian tetrapods. In particular, hind limbs
with a complete array of digits are notably rare. The unusual construction of the pes of
*Termonerpeton* prompted us to examine the hind limb morphology of all East Kirkton
tetrapods and other Carboniferous taxa (Fig 4). We compared the phalangeal formula and
digit length and proportions of *Termonerpeton* with those of six other named East Kirkton
taxa (Fig 4 a-g; the limbless *Ophiderpeton* was obviously excluded). East Kirkton tetrapods
show a remarkable variety in limb morphology and proportions that bear comparison with
those of later Carboniferous and early Permian taxa.

We illustrate the epipodials, tarsi, and digits of seven East Kirkton taxa alongside
those of a range of other tetrapods for comparison (Fig. 4). All are drawn to a common tibial
length, except for the amniote *Petrolacosaurus*, in which the epipodials are greatly elongate.

Figure 4 here

In terms of the of pes size relative to the tibia, the East Kirkton taxa *Balanerpeton*,
*Eucritta*, and *Silvanerpeton* (Fig. 4 a, b, d) are similarly proportioned, whereas *Eldeceeon* and
*Westlothiana* (Fig. 4 c, e) exhibit somewhat larger pedes. *Kirktonecta* has proportionally the
largest pedes of all (Fig. 4 f). *Termonerpeton* (Fig. 4 g) has a pes of similar size to the first
three taxa except that digit IV is relatively much longer than in any of the others, with an

exceptionally large metatarsal IV. In all those taxa in which digit IV is fully preserved, it is
the longest, especially in *Eldeceeon* and *Kirktonecta*, but in none does it approach in size and
proportions that of *Termonerpeton*.

The illustrated limbs also differ from one another in the degree of ossification of the
tarsal bones. Most taxa except *Eucritta* have some indication of ossified tarsal elements, and
several of them show a complete or almost complete set. *Kirktonecta* does have an ossified
tarsus, but the preservation of the specimen does not allow us to identify individual elements.

The phalangeal count, where known, also varies: 22343 in *Balanerpeton* (2Milner &
Sequeira 1994); 223?? in *Eucritta* (12Clack 1998); 23455 in *Silvanerpeton* (4Ruta & Clack
2002); 23454 in *Eldeceeon*, *Kirktonecta*, *Termonerpeton*, and *Westlothiana*.

This range of morphology is evidence that the earliest known terrestrial tetrapods
were experimenting with limb and digit proportions, as well as phalangeal constructions,
probably in response to the varied terrain of the terrestrial environment. All share a five-digit
pes, a condition that is first recorded in the Tournaisian (24Clack and Finney 2005), and there
is no evidence of polydactyly in any other known Carboniferous tetrapod, such as is found in
late Devonian taxa (e.g. *Acanthostega*; *Ichthyostega*; *Tulerpeton*; 25Clack 2012).

We also compared the pedes of the East Kirkton taxa with those of seven other
Palaeozoic genera (Fig. 4 h-n): one earlier, *Pederpes* (24Clack & Finney 2005); one almost
contemporary, *Caerorhachis* (26Ruta, Milner & Coates 2002); four later Carboniferous,
*Greererpeton* (27Godfrey 1989), *Hylonomus* (28Carroll 1964), *Tuditanus* (22Carroll &
Gaskill 1978), and *Petrolacosaurus* (29Reisz 1981); and one early Permian, *Archeria*
(30Romer 1957). Of these, *Greererpeton* has relatively the smallest pes. In most, digit IV is
the longest, though in *Pederpes* and *Caerorhachis* it is incomplete. The pes of *Caerorhachis*
was originally restored (31Holmes & Carroll 1977) with only three phalanges in digit IV.
This is probably incorrect and would be unusual in Carboniferous tetrapods. The pes of the

anthracosaur *Archeria* was originally reconstructed as having digit V as the longest (30Romer
1957), but again this is unusual among later Carboniferous and early Permian tetrapods and
we suspect that digits IV and V have been transposed in the reconstruction of the *Archeria*
pes. Romer himself expressed doubt about this reconstruction. In either case, its phalangeal
formula is similar to that of the East Kirkton anthracosaur *Silvanerpeton*, as 23455.

Among Carboniferous tetrapods, temnospondyls such as *Balanerpeton* and colosteids
such as *Greererpeton* show a digit IV that is somewhat longer than the others, but metatarsal
IV is very similar in length and breadth to the adjacent metatarsals. In anthracosaurs, digit IV
is the longest, but again metatarsal IV is not significantly broader than adjacent metatarsals.
This is also the case in the early amniote *Hylonomus* and the microsauro *Tuditanus*. Among
the taxa illustrated here, *Termonerpeton* shows a strikingly similar pes to that of the Late
Pennsylvanian araeoscelidian diapsid *Petrolacosaurus* (Fig. 4 n). In both, metatarsal IV is
significantly longer and stouter than others in the same pes, and forms part of a similarly long
digit IV. In early amniotes, an elongate digit IV coupled with an elongate metatarsal IV is a
common occurrence in other taxa, such as protothyridids (e.g. *Anthracodromeus* 32Carroll &
Baird 1972), basal araeoscelidians (e.g. *Spinoaequalis* 33deBraga & Reisz 1995), younginids
(e.g. *Youngina* 34Smith & Evans 1996), saurians (34Lee 1997), and basal synapsids (e.g.
*Heleosaurus* 35Carroll 1976, 36Reisz & Modesto 2007), among others.

However, an elongate metatarsal IV and associated digit are not universal among
Palaeozoic amniotes. In the eureptile captorhinid *Eocaptorhinus*, digit IV is also the longest,
but the length of metatarsal IV does not greatly exceed that of other metatarsals (38Heaton &
Reisz1980). The same is true of some early Permian stem amniotes such as seymouriamorphs
(e.g. *Seymouria* 39White 1939; *Discosauriscus* 40Klembara and Bartik 1999), and diadectids
(e.g. *Diadectes* 41Berman & Henrici 2003) although in *Orobates* digit III is a little longer
than digit IV (42Berman *et al.* 2004). Among synapsids, dicynodonts such as *Diictodon*

(43Ray & Chinsamy 2003) and caseids (44Stovall *et al.* 1966) all have five pedal digits of
more or less uniform length.

Among modern lizards, elongate metatarsal and digit IV are essentially universal
features and often specialized. In runners, they are probably associated with increased stride
length (45Irschick & Jayne 1999). Digit IV is the last to leave the ground during the stride
cycle and provides extra leverage during running. Elongate digits also assist energy-saving in
rapid locomotion because the tendons act as springs (46Li *et al.* 2012). An elongate digit IV
is known to assist locomotion on a range of surfaces in the Zebra-tailed lizard (46Li *et al.*
2012) and may have played a similar role in a complex environment like that of East Kirkton.

An elongate digit IV may be primitive for amniotes, being present in *Hylonomus*,
*Paleothyris*, and *Petrolacosaurus* (Fig 4 l, n), and shortening of this digit may represent a
derived feature. In later amniotes, the conditions vary, with larger, heavier-bodied tetrapods
such as dicynodonts and diadectids having generally shorter toes and adopting a more clearly
plantigrade posture. Some of these features have recently been recognised in fossil footprints
from the late Carboniferous and Permian (47Lucas *et al.* 2020, 48Buchwitz *et al.* 2021).
However, given the range of hind limb morphologies present at East Kirkton, and the
amniote-like pes found in *Termonerpeton*, we would advise caution when ascribing particular
footprints to specific clades.

Figure 5 here

**Phylogenetic relations of *Termonerpeton*.** The major implication of our findings is that
*Termonerpeton* is the earliest known tetrapod to exhibit a fundamentally amniote-like pes
construction. The results of our phylogenetic analyses lend some support to the interpretation
of this taxon as a stem amniote, despite its uncertain placement in the unweighted-character

parsimony analysis (Fig. 5 a). All other analyses – implied weights, reweighted characters,
 and Bayesian – place *Termonerpeton* on the amniote stem, albeit in different positions. In the
 implied weights analysis (Fig. 5 b), *Termonerpeton*, *Silvanerpeton*, and *Eldeceeon* form a
 clade placed immediately crownward of chroniosaurs plus anthracosaurs and anti-crownward
 of paraphyletic gephyrostegids. In the reweighted analysis, (Fig. 5 c) *Termonerpeton* and
 *Caerorhachis* appear as successive sister taxa, in that order, to a clade of anthracosaurs. In the
 Bayesian analysis (Fig. 5 d), the amniote total group receives moderate support (76), with
 *Caerorhachis* as the most plesiomorphic stem amniote taxon. Crownward of *Caerorhachis* is
 a polytomous node with low support (59), subtending *Termonerpeton*, a clade of *Eldeceeon*
 and *Silvanerpeton*, a clade of anthracosaurs, and a clade consisting of all remaining taxa. In
 crownward sequence, the latter shows chroniosuchians, gephyrostegids, seymouriamorphs,
 *Solenodonsaurus*, and *Westlothiana* as successive sister groups to a strongly supported (100)
 clade formed of synapsids, eureptiles diadectomorphs. Eureptile monophyly is not retrieved,
 but strong support (100) is assigned to the branch subtending diadectomorphs plus synapsids.

Figure legends

**Figure 1. *Termonerpeton mackrydactylus* gen. et sp. nov. holotype UMZC 2019.1. a,**
 **specimen photograph. b, interpretive drawing. Scale bars 10 mm. Abbreviations: ha, haemal**
 **arch; l, left; na, neural arch; phal, phalange; plc, pleurocentrum; posac, postsacral; presac,**
 **presacral; r, right; sac, sacral.**

**Figure 2. *Termonerpeton mackrydactylus* gen. et sp. nov. left hind limb of UMZC 2019.1.**
 **a, specimen photograph, b, interpretive drawing, centralia labelled in red, distal tarsals**
 **labelled in blue, metatarsals labelled in black, c, reconstruction of left tibia, fibula and pes.**
 **Scale bars 10 mm.**

**Figure 3. *Termonerpeton mackrydactylus* gen. et sp. nov partial pes, attributed specimen**
**NMS G 1992.22.1. a**, specimen photograph, **b**, interpretive drawing, centralia labelled in red,
distal tarsals labelled in blue, metatarsals labelled in black. Scale bars 10 mm.

**Figure 4. Comparison of the left tibia, fibula, tarsus, and digits of early tetrapods. a**,
*Balanerpeton* after 2, **b**, *Eucritta* after 12, **c**, *Eldeceeon* after 6, **d**, *Silvanerpeton* after 4, **e**,
*Westlothiana* after 7, **f**, *Kirktonecta* original, see 15, **g**, *Termonerpeton*, **h**, *Pederpes* after 24,
**i**, *Greererpeton* after 27, **j**, *Caerorhachis* after 31, **k**, *Archeria* after 30, **l**, *Hylonomus* after 28,
**m**, *Tuditanus* after 22, **n**, *Petrolacosaurus* after 29. Drawn to the same tibial length apart
from **n**. Scale bars 10 mm.

**Figure 5. Results of phylogenetic analyses. a**, strict consensus of 120 shortest trees from
unweighted analysis (tree length = 1286 steps, ensemble consistency index C.I. = 0.2738
without uninformative characters, ensemble retention index R.I. = 0.5768), **b**, single tree
from implied weights analysis (tree length = 1298 steps, Goloboff fit = -202.59266, C.I. =
0.2712, R.I. = 0.5713), **c**, single tree from reweighted analysis (tree length = 212,68965 steps,
C.I. = 0.4755, R.I. = 0.774), **d**, Bayesian topology with branches reporting credibility values.

Methods

**Preparation and visualization.** The specimen was prepared using an Emax Evolution Grinder
and Polisher fitted with a chuck to take a tungsten carbide rod. Photography by JAC used a
Panasonic Lumix DMC-LZ5, specimen drawings by JAC were made using a camera lucida,
all followed by processing with Photoshop CC 2017 or 2019.

**Phylogenetic analysis.** In order to evaluate the phylogenetic position of *Termonerpeton*, we
employed a slightly expanded version of the taxon-character data matrix in Klembara et al.
(49 2020) and subjected it to maximum parsimony and Bayesian inference analyses. Prior to
analyses, the data matrix was inspected for possible occurrences of ‘rogue’ species (*sensu* 50

Wilkinson 1996) causing loss of resolution among more stable species. No ‘rogue’ taxa were
detected using the ‘safe_taxonomic_reduction’ function of the package Claddis (51 Lloyd
2016) in the R (v. 4.0.3) environment for statistical computing (<https://cran.r-project.org>). In
the case of parsimony analyses, we explored the results obtained with unweighted, posteriorly
weighted and standard implied weighted characters (52 Goloboff 1993). Posterior weighting
employed the maximum value (best fit) of the rescaled consistency index of each character,
such as was obtained from the unweighted analysis, whereas implied weighting used a value
of 6 for the constant of concavity K. All parsimony analyses were performed in PAUP* v.
4.0a build 169 (53 Swofford 1998; <https://paup.phylosolutions.com>) using identical settings,
as follows: heuristic search method; tree bisection-reconnection branch-swapping algorithm
with 10,000 random stepwise taxon addition sequences, holding one tree in memory at each
replicate; 10 consecutive rounds of branch-swapping applied to all trees in memory from this
initial search, but with multiple trees saving option in effect. Node support was evaluated
with bootstrap (54 Felsenstein 1985) and jackknife (55 Farris *et al.* 1996), in each case with
10,000 random character resampling replicates using the fast stepwise addition option.

The Bayesian inference analysis was carried out with MrBayes v. 3.2.6 (56 Ronquist
and Huelsenbeck 2003) using the standard data type option for morphological characters, and
an equal-rate model of character-state change. The analysis employed four chains with 10^7
generations, sampling every 1000 generations, and discarding 25% of the obtained samples.
At the end of the search, we saved branch lengths and clade credibility values. Convergence
was regarded as satisfactory based upon the values of the Potential Scale Reduction Factor
(57 Gelman and Rubin 1992) approaching 1.

**Data availability**

The character list and data matrix used in this study are available in Supplementary Data 1–2.

**References**

- 1. Mann, A., Gee, B. M., Pardo, J. D., Marjanovic, D., Adams, G. R., Calthorpe, A. S.,
Maddin, H. C. and Anderson, J. S. Reassessment of historic ‘microsaurs’ from Joggins,
Nova Scotia, reveals hidden diversity in the earliest amniote ecosystem. *Papers in*
*Palaeontology* **6**, 605–625. (2020)
- 2. Milner, A. R. and Sequeira, S. E. K. The temnospondyl amphibians from the Viséan of
East Kirkton, West Lothian, Scotland. *Transactions of the Royal Society of Edinburgh:*
*Earth Sciences* **84**, 331–361. (1984)
- 3. Clack, J. A. *Silvanerpeton miripedes*, a new anthracosauroid from the Viséan of East
Kirkton, West Lothian, Scotland. *Transactions of the Royal Society of Edinburgh:*
*Earth Sciences* **84**, 369–376. (1994)
- 4. Ruta, M and Clack, J. A. A review of *Silvanerpeton miripedes*, a stem amniote from the
Lower Carboniferous of East Kirkton, West Lothian, Scotland. *Transactions of the*
*Royal Society of Edinburgh: Earth Sciences* **97**, 31–63. (2006)
- 5. Smithson, T. R. *Eldeceeon rolfei*, a new reptiliomorph from the Viséan of East Kirkton,
West Lothian, Scotland. *Transactions of the Royal Society of Edinburgh: Earth*
*Sciences* **84**, 377–382. (1994)
- 6. Ruta, M., Smithson, T. R. and Clack, J. A. A review of the stem amniote *Eldeceeon*
*rolfei*, from the Visean of East Kirkton. *Earth and Environmental Science Transactions*
*of the Royal Society of Edinburgh.* **111**, 173-192. (2020)
- 7. Smithson, T. R., Carroll, R. L., Panchen, A. L. and Andrews, S. M. *Westlothiana lizziae*
from the Viséan of East Kirkton, West Lothian, Scotland, and the amniote stem.
*Transactions of the Royal Society of Edinburgh: Earth Sciences* **84**, 383–412. (1994)

- 8. Clack, J. A. and all Bennett, C. E., Carpenter, D. K., Davies, S. J., Fraser, N. C.,
Kearsley, T. I., Marshall, J. E. A., Millward, D., Otoo, B. K. A., Reeves, E. J., Ross, A.
376 J., Ruta, M., Smithson, K. Z., Smithson, T. R. & Walsh, S. A. 2016. Phylogenetic and
377 environmental context of a Tournaisian tetrapod fauna. *Nature Ecology and Evolution* **1**
(s41559), 1–11. (2016)
- 9. Clack, J. A. Ruta, M., Milner, A. R., Marshall, J. E. A., Smithson, T. R. & Smithson, K.
Z. *Acherontiscus caledoniae*: the earliest heterodont and durophagous tetrapod. *Royal*
*Society Open Science* **6 (182087)**, 1–10. (2019)
- 10. Pardo, J. D., Szostakiwskyj, M., Ahlberg, P. E. & Anderson, J. S. Hidden
morphological diversity among early tetrapods. *Nature* **546**, 642–645. (2017)
- 11. Marjanovic, D. and Laurin, M. Phylogeny of Paleozoic limbed vertebrates reassessed
through revision and expansion of the largest published relevant data matrix. *PeerJ* **6**
(e5565), 1–191. (2019)
- 12. Clack, J. A. A new Early Carboniferous tetrapod with a *mélange* of crown-group
characters. *Nature*. **394**, 66–9. (1998)
- 13. Clack, J. A. *Eucritta melanolimnetes* from the Early Carboniferous of Scotland, a stem
tetrapod showing a mosaic of characteristics. *Transactions of the Royal Society of*
*Edinburgh: Earth Sciences* **92**, 75–95. (2001)
- 14. Milner, A. C. The aïstopod amphibian from the Viséan of East Kirkton, West Lothian,
Scotland. *Transactions of the Royal Society of Edinburgh: Earth Sciences* **84**, 363–368.
(1994)
- 15. Clack, J. A. A new microsauro from the early Carboniferous (Viséan) of East Kirkton,
Scotland, showing soft tissue evidence. *Special Papers in Palaeontology* **86**, 1–11.
(2011)

- 16. Sues, H-D. Authorship and date of publication of the name Tetrapoda. *Journal of*
*Vertebrate Paleontology* **39**, e1564758. (2019)
- 17. Smithson, T. R. Scottish Carboniferous amphibian localities. *Scottish Journal of*
*Geology*. **21**, 123–142. (1985)
- 18. Smithson, T. R. and Rolfe, W. D. I. What made Stan Wood a great collector? *Earth and*
*Environmental Science Transactions of the Royal Society of Edinburgh* **108**, 7–17.
(2018)
- 19. Rolfe, W. D. I., Durrant, G. P., Baird, W. J., Chaplin, C., Paton, R. L. and Reekie, R. J.
The East Kirkton Limestone, Viséan of Westlothian, Scotland: introduction and
stratigraphy. *Transactions of the Royal Society of Edinburgh: Earth Sciences*. **84**, 177–
188. (1994)
- 20. Rolfe, W. D. I. Clarkson, E. N. K & Panchen, A. L. (eds) *Volcanism and early*
*terrestrial biota*. *Transactions of the Royal Society of Edinburgh: Earth Sciences* **84**
(Parts 3 & 4). Edinburgh: The Royal Society of Edinburgh. 467 pp. (1994)
- 21. Clack, J. A. The East Kirkton Lagerstätte: a window onto Early Carboniferous land
ecosystems. In Fraser, N. C & Sues, H. D. (eds) *Terrestrial conservation lagerstätten:*
*windows into the evolution of life on land*, 39–64. Edinburgh: Dunedin Academic
Press. (2017)
- 22. Carroll, R. L. and Gaskill, P. The Order Microsauria. *Memoirs of the American*
*Philosophical Society* **126**, 1–211. (1978)
- 23. Holmes, R. B. The Carboniferous amphibian *Proterogyrinus scheelei* Romer, and the
early evolution of tetrapods. *Philosophical Transactions of the Royal Society of*
*London, Series B* **306**, 431–527. (1984)

- 24. Clack, J. A. and Finney, S. M. *Pederpes finneyae*, an articulated tetrapod from the
Tournaisian of Western Scotland. *Journal of Systematic Palaeontology* **2**, 311–346.
(2005)
- 25. Clack, J. A. *Gaining ground: the origin and evolution of tetrapods*. Second Edition.
Bloomington: Indiana University Press. 523 pp. (2012)
- 26. Ruta, M., Milner, A. R. and Coates, M. I. The tetrapod *Caerorhachis bairdi* Holmes
and Carroll from the Lower Carboniferous of Scotland. *Transactions of the Royal*
*Society of Edinburgh: Earth Sciences* **92**, 229–261. (2002)
- 27. Godfrey, S. The postcranial skeletal anatomy of the Carboniferous tetrapod
*Greererpeton burkemorani*. *Philosophical Transactions of the Royal Society of London*
*Series B* **323**, 75–133. (1989)
- 28. Carroll, R. L. The earliest reptiles. *Journal of the Linnean Society (Zoology)* **45**, 61–83.
(1964)
- 29. Reisz, R. R. A diapsid reptile from the Pennsylvanian of Kansas. *Special Publication of*
*the Museum of Natural History, University of Kansas* **7**, 1–74. (1981)
- 30. Romer, A. S. The appendicular skeleton of the Permian embolomere amphibian
*Archeria*. *Contributions from the Museum of Paleontology, University of Michigan* **13**,
103–159. (1957)
- 31. Holmes, R. B. and Carroll, R. L. A temnospondyl amphibian from the Mississippian of
Scotland. *Bulletin of the Museum of Comparative Zoology, Harvard College* **147**, 489–
511. (1977)
- 32. Carroll, R. L. and Baird, D. Carboniferous stem-reptiles of the Family Romeriidae.
*Bulletin of the Museum of Comparative Zoology, Harvard College* **143**, 321–364.
(1972)

- 33. de Braga, M. and Reisz, R. R. A new diapsid reptile from the uppermost
Carboniferous (Stephanian) of Kansas. *Palaeontology* **38**, 199–212. (1995)
- 34. Smith, R. M. H. and Evans, S. E. New material of Youngina: evidence of juvenile
aggregation in Permian diapsid reptiles. *Palaeontology* **39**, 289–303. (1996)
- 35. Lee, M. S. Y. The evolution of the reptilian hindfoot and the homology of the hooked
fifth metatarsal. *Journal of Evolutionary Biology* **10**, 253–263. (1997)
- 36. Carroll, R. L. Eosuchians and the origin of archosaurs. In C. S. Churcher (ed.) Athlon:
Essays on Paleontology in Honour of Loris Shano Russell. *Miscellaneous Publications*
*of the Royal Ontario Museum, Toronto* 58–76. (1976)
- 37. Reisz, R. R. and Modesto, S. P. *Heleosaurus scholtzi* from the Permian of South
Africa: a varanopid synapsid, not a diapsid reptile. *Journal of Vertebrate Paleontology*
**27**, 734–739. (2007)
- 38. Heaton, M. and Reisz, R. R. A skeletal reconstruction of the early Permian captorhinid
reptile *Eocaptorhinus laticeps* Williston. *Journal of Paleontology* **54**, 136–143. (1986)
- 39. White, T. E. Osteology of *Seymouria baylorensis* Broili. *Bulletin of the Museum of*
*Comparative Zoology* **85**, 325–409. (1939)
- 40. Klembara, J. and Bartik, I. The postcranial skeleton of *Discosauriscus* Kuhn, a
seymouriamorph tetrapod from the Lower Permian of the Boskovice Furrow (Czech
Republic). *Transactions of the Royal Society of Edinburgh: Earth Sciences* **90**, 287–
316. (1999)
- 41. Berman, D. S. and Henrici, A. C. Homology of the astragalus and structure and
function of the tarsus of Diadectidae. *Journal of Paleontology* **77**, 172–188. (2003)
- 42. Berman, D. S., Henrici, A. C., Kissel, R. A., Sumida, S. S. and Martens, T. A new
diadectid (Diadectomorpha), *Orobates pabsti*, from the Early Permian of central
Germany. *Bull. Carnegie Mus. Nat. Hist.* **35**, 1–36. (2004)

- 43. Ray, S. and Chinsamy, A. Functional aspects of the postcranial anatomy of the Permian
dicynodont *Diictodon* and their ecological implications. *Palaeontology* **46**, 151–187.
(2003)
- 44. Stoval, J. W., Price, L. I. and Romer, A. S. The postcranial skeleton of the giant
Permian pelycosaur *Cotylorhynchus romeri*. *Bulletin of the Museum of Comparative*
*Zoology, Harvard College* **135**, 1–30. (1966)
- 45. Irschick, D. J. and Jayne, B. C. Comparative three-dimensional kinematics of the
hindlimb for high-speed bipedal and quadrupedal locomotion of lizards. *Journal of*
*Experimental Biology* **202**, 1047–1065. (1999)
- 46. Li, C., Tonia Hsieh, S. and Goldman, D. I. Multi-functional foot use during running in
the zebra-tailed lizard (*Callisaurus draconoides*). *Journal of Experimental Biology*
**215**, 3293–3308. (2012)
- 47. Lucas, S. G., Stimson, M. R., King, O. A., Calder, J. H., Mansky, C. F., Herbert, B. L.
and Hunt, A. P. Carboniferous tetrapod footprint biostratigraphy, biochronology and
evolutionary events. *Geological Society, London, Special Papers*.
DOI:<https://doi.org/10.1144/SP512-2020-235>.
- 48. Buchwitz, M., Jansen, M., Renaudie, J., Marchetti, L. and Voigt, S. Evolutionary
change in locomotion close to the origin of amniotes inferred from trackway data in an
ancestral state reconstruction approach. *Frontiers of Ecology and Evolution* **9**:647779
(2021)
- 49. Klembara, J., Hain, M., Ruta, M., Berman, D. S., Pierce, S. E. and Henrici, A. C. Inner
ear morphology of diadectomorphs and seymouriamorphs (Tetrapoda) uncovered by
high-resolution x-ray microcomputed tomography and the origin of the amniote crown-
group. *Palaeontology* **63**, 131-154. (2020)

- 50. Wilkinson, M. Majority-rule consensus trees and their use in bootstrapping. *Molecular*
*Biology and Evolution* **13**, 437-444. (1996)
- 51. Lloyd, G. T. Estimating morphological diversity and tempo with discrete character-
taxon matrices: implementation, challenges, progress and future directions. *Biological*
*Journal of the Linnean Society* **118**, 131-151. (2016)
- 52. Goloboff, P. Estimating character weighting during tree search. *Cladistics* **9**, 83-91.
(1993)
- 53. Swofford, D. L. *PAUP* Phylogenetic analysis using parsimony (*and other methods).*
*Version 4*. Sunderland, Massachusetts: Sinauer Associates. (1998)
- 54. Felsenstein, J. Confidence limits on phylogenies: An approach using the bootstrap.
*Evolution* **39**, 783-791. (1985)
- 55. Farris, J. S., Albert, V. A., Källersjö, M., Lipscomb, D. and Kluge, A. G. Parsimony
jackknifing outperforms neighbour-joining. *Cladistics* **12**, 99-124. (1996)
- 56. Ronquist, F. and Huelsenbeck, J. P. MRBAYES 3: Bayesian phylogenetic inference
under mixed models. *Bioinformatics* **19**, 1572-1574. (2003)
- 57. Gelman, A. and Rubin, D. B. Inference from iterative simulation using multiple
sequences. *Statistical Science* **7**, 457-472. (1992)

Acknowledgements

We thank Daniel Delbarre who prepared the UMZC specimen during an undergraduate
summer project and Sarah Wallace-Johnson (formerly Finney) from the Earth Sciences
Department, University of Cambridge, for the loan of the Emax Evolution Grinder and
Polisher. We are grateful to Matt Lowe of the University Museum of Zoology Cambridge for
access to UMZC 2019.1 and Nick Fraser and Stig Walsh of National Museums Scotland for
the loan of NMS G 1992.22.1.

**Authors contributions**

JAC conceived the study. JAC and TRS described and analysed the tetrapod specimen and
prepared the figures. MR undertook the phylogenetic analyses. All authors interpreted the
data and wrote the manuscript.

**Competing interests**

The authors declare no competing interests.

10 mm

r pelvis

gastralia

na

plc

ic

na

gastralia

posac rib

na

sac rib

posac rib

phal

iliac blade

l pelvis

na

trunk rib

ha

presac rib

phal

plc

ha

ic

na

c

b

Rebuttal Letter

Reviewer #1 (Remarks to the Author):

This manuscript described a new genus of early tetrapods from the late Mississippian East Kirton Limestone, increasing the taxonomic diversity and morphological disparity of a well-known Carboniferous terrestrial fauna. The referred fossils are well described and illustrated, and the discussions and comparisons with other related taxa are convincing. As such, I recommend its publication in *Communications Biology*.

We thank Reviewer 1 for their kind words and very supportive remarks.

However, I have following minor comments for consideration.

1) Being the earliest known tetrapod to exhibit a fundamentally amniote-like pes construction, the new form, however, cannot be definitely referred to as a stem-amniote, as shown by the parsimony analysis. This point has not been clearly stated in the abstract.

The reviewer is correct. We have clarified this point and fine-tuned our statement about the affinities of the new taxon given the results of the unweighted parsimony analysis. We are in the process of preparing additional work on other East Kirton tetrapods and, as part of this, we are expanding and refining the taxon-character databases that underpin the phylogenetic analyses. Although we are not in a position to pre-empt our own research efforts in this area, we can anticipate that preliminary results allow us to place our new taxon more firmly among stem amniotes, largely in agreement with some of the results obtained from our other analyses (Bayesian and weighted character parsimony).

2) Differential diagnosis might be better than the present diagnosis. Which characters separate the new taxon from other related tetrapods?

We thank the reviewer for this suggestion. A differential diagnosis now features in the revision of our manuscript. It starts with autapomorphic (or presumed autapomorphic) conditions and ends with characters for which polarity is uncertain at present. We have attempted to distill unique features first, before presenting information on putative traits shared at different levels of the early tetrapod taxonomy.

3) The reference citations in the main-text and figures need careful edition.

We have vetted the whole reference list, removed authors' names from the main text (which we had placed in the original submission for the purpose of easy reference retrieval), and reformatted the reference style.

Reviewer #2 (Remarks to the Author):

The manuscript describes a new tetrapod from the Early Carboniferous period and uses it as the basis for a new analysis of the early evolution of the tetrapod pes. This is an important discovery because of the relative lack of tetrapod fossil material from this time period, when tetrapods first became terrestrially capable and the amniote and amphibian lineages diverged. The pes of *Termonerpeton* is notable for being the earliest example of "amniote-like construction," mainly due to an elongate 4th digit.

We thank Reviewer 2 for their appreciative comments and very helpful suggestions.

The manuscript is well written with high quality figures. It describes an important discovery that bears on the evolution of terrestrial locomotion in tetrapods. The impact might be enhanced by a more focused discussion (specific comments below).

We endorse this remark in full and we have amended, reorganized, and expanded the discussion. We talk about the importance of the new taxon at greater length than in the original version, expanding on morphofunctional and evolutionary implications of its anatomy.

Specific comments

In the results section it is sometimes unclear whether the dorsal or ventral aspect of a bone is being described.

We have rectified this by adding remarks in appropriate places, especially in the description of the pelvis and individual limb bones.

Line 98 refers to a probable acetabulum; please label in Figure 1b and describe if possible.

We have labelled the acetabulum, but kept its description to a minimum, for obvious reasons: the area corresponding to the acetabular region appears as an indistinct depression and not much of its morphology can be discerned.

Line 125: "the proximal end [of the femur] also appears concave" i.e., the ventral aspect of the femoral head? The proximal end looks convex in the photo and illustration. Please clarify.

We clarify that we are referring to a subcentral depressed area on the extensor surface of the proximal extremity of the bone.

The discussion section focuses mainly on the pes. In general, I would appreciate more explicit discussion of the relevance of these characters to the issues brought up in the introduction, namely early appearance of certain hindlimb specializations, underappreciated diversity among the tetrapods of the East Kirkton locality, and terrestrial adaptations.

We entirely agree. We have rewritten the discussion to accommodate the reviewer's remarks. In the revised version, the treatment of pes morphology is grafted onto a wider discussion that tackles possible functional adaptations of the new taxon,

especially in terms of its possible locomotory habits, and the evolutionary implications of its discovery.

Lines 212-213: "experimenting ... probably in response to the varied terrain of the terrestrial environment" - because it was new to them or because terrestrial environments are intrinsically more varied than aquatic ones? Or is the East Kirkton environment particularly variable?

We clarify that the East Kirkton terrain is highly variable. We have shifted the relevant part of text to the new, expanded discussion, so as to provide continuity of argument exposition.

What was the purpose of comparing the pedal proportions with that of other Carboniferous tetrapods? This section seems unfocused. I suggest re-organizing it around specific predictions related to ecology, phylogeny, etc.

We think a comparison of the hind limb morphology of the new taxon with those of its contemporaries from East Kirkton is pivotal. We emphasize the uniqueness of the new taxon by drawing similarities and differences with its contemporaries. No such detailed comparisons have been made in previous accounts of the fauna from the site. The comparisons among hind limb morphologies appear to us well aligned with the new discussion. If possible, we would like to keep the comparisons section within the results, such that the discussion is solely targeted at the possible functional roles of the new taxon's pes and the evolutionary implications of its discovery.

Lines 253-259: I am happy to see a consideration of locomotor function of D4. However, this paragraph is vaguely worded and the point is unclear. If an elongate MT4 and D4 are "nearly universal" in modern lizards, how can they be associated with different functional characteristics?

A proper consideration of digit IV elongation is given in the revised discussion, particularly in terms of its possible functions. Please see also previous remarks. We note that, even within clades of lizards in which digit IV maintains similar proportions in relation to the other digits, possible alternative functions are possible. And while functional inference in the new taxon necessarily entails elements of speculation, we think it possible to make reasonable inference (e.g. see our short new section detailing why the pes of the new taxon may have acted as a load-bearing device and why it may not have been a cursorial animal.)

Presumably it has to do with extreme elongation, but how is that defined?

Indeed, it is difficult to characterize digit elongation. We have tried to reword this section. However, we do clarify in the text what is unique about the digit elongation in the new taxon. It is a combination of enlargement of metatarsal IV and the fact that the corresponding digit IV is distinctively longer than the adjacent ones, and is also much more robust. No such combination of traits is found in other fossil tetrapods, as far as we can tell.

Why is varied substrate the preferred functional hypothesis for its presence in Termonerpeton - because it doesn't aid in stride length or leverage during slow locomotion?

We have clarified our meaning in the relevant section of the discussion. It may well be that, as in some modern lizards, an elongate pes affords the animal the ability to locomote on diverse substrates. We suggest, albeit tentatively, that a spring mechanism linked to tendons of digit IV may have operated in the new taxon in a similar fashion to what some extant lizards achieve. Specifically, it would facilitate leverage during walking while saving energy. A long foot, as in the contemporary *Eldeceeon*, may have promoted increase in stride length without necessarily involving higher frequency of muscle contraction – again, an energy-saving mechanism that would also permit increased speed.

Lines 273-274: "fundamentally amniote-like pes construction" meaning 5 toes and elongate MT4/D4?

Indeed, this is what we had in mind, and we have clarified this in the refined discussion, under 'Evolutionary implications'. Tarsal construction (specifically, enlarged proximal moiety) is also key here and we have added relevant sections in the text

I suggest a few sentences at the end of the discussion summing up the big-picture implications of the discovery. For example, to me the morphology of this fossil seems to suggest that adaptations for more versatile/advanced terrestrial locomotion were present at the very base of the amniote lineage. I'd love to hear what the authors feel is the most important takeaway or most interesting possibility raised by this exciting discovery.

We agree with the reviewer, and we have produced a new version of the discussion in which we have articulated the implications of our findings. We have tried to make reasonable inferences as to the possible locomotory mode of the new taxon, especially in relation to its pes morphology. In addition, we have provided a pithy summary of the evolutionary implications of the new discovery, particularly taking into account the alternative phylogenetic placements of the new taxon among stem amniotes.

Reviewer #3 (Remarks to the Author):

I generally encourage publication of the described specimens because of our still limited understanding of early tetrapod evolution, but I have some major points of criticism that need to be addressed.

We thank Reviewer 3 for appreciating the relevance of the new taxon in the context of early tetrapod evolution and for his incisive comments. We have endeavoured to respond to his criticism of certain portions of the text and iconography.

1) the quality and labeling of the figures, especially of Fig. 1 and 2 need to be improved

This has been rectified. We have produced photographs with enhanced contrast and re-labelled the line tracings of the specimens.

2) all elements of the tarsus need to be clearly identified and labelled because much of the discussion is centered around these.

We have produced better and sharper labels for the line tracings of the specimens. We have also used new and sharper photographs that, we hope, do more justice to the specimen morphology. East Kirkton material is notoriously difficult, but observations of fine details are certainly possible, and we have been able to refine our own observations on countless instances in the course of our individual and collaborative efforts.

3) In my opinion, the interpretation of much of the pes is rather speculative, and other bones are damaged or crushed, thus providing limited information. The authors should reassess whether the material is sufficient for describing a new species and genus.

We fully respect the reviewer's stance, but we emphatically disagree that the interpretation of much of the pes in the new taxon is rather speculative. The interpretation of the new taxon is not without difficulties, but we remain strongly confident that most morphological details can be gleaned from it. To aid in the 'reading' of the disarticulated, but otherwise nearly complete pes of the holotype, we have traced the 'paths' along phalanges that appear in close proximity to one another and the arrangement of which, in our opinion, is fully consistent with a retracted pes with flexed digits. While there is some damage, the outline of the postcranial elements can be discerned. Furthermore, what is preserved allows us to rule out any of the other tetrapods from East Kirkton in terms of species assignment or, for that matter, any later Paleozoic tetrapods that we know of. Even without a pes, the uniqueness of the new taxon can be supported by its pelvis morphology and, to a degree, the combination of other traits, such as vertebral centra and ribs. Our thoughts on this matter are crystallized in a new, revised, differential diagnosis. Finally, the presence of a referred specimen allows us to build our argument more confidently as data from both specimens are reciprocally illuminating.

4) Additional images (with angled/side light) or coating with ammonium chloride should be considered - not mandatory though.

We agree with the reviewer, and we have produced better figures. We did not use ammonium chloride sublimate as this is known to affect, long-term, the quality of fossil material.

I have made other comments directly on the PDF.

We have amended sections of text where the reviewer indicated on the pdf of our original submission. Below is a list of amendments that follows the progression of annotations supplied by the reviewer.

1) We have added 'early Carboniferous' to the title.

- 2) Unlike in our previous version, we have followed the practice of not introducing new genera and species in the abstract. We hope this is satisfactory.
- 3) We have clarified what alternative interpretation have been proposed for the affinities of East Kirkton tetrapods.
- 4) We have replaced the phrase 'plesiomorphic members' with a new line of text that reflects closely the reviewer's suggestion.
- 5) We have replaced 'disrupted' with 'disarticulated' in the description of the holotype under the systematic palaeontology section.
- 6) In the new, differential diagnosis we have retained digit proportions in the list of diagnostic features. As explained above, the preservation of the holotype and the morphology of the referred specimen leave no doubt as to the proportions of metatarsal IV and the elongation of the digits, particularly digit IV. We hope the revised figures make the details a little clearer. Ultimately, the specimen is the sole bearer of information and is there for any interested researcher to scrutinise.
- 7) We have rebuilt our images using sharper photographs and relabelled the diagrams. We hope this is satisfactory and meets the requests from the referee.

REVIEWERS' COMMENTS:

Reviewer #2 (Remarks to the Author):

The authors have addressed all of my major concerns. I am very happy with this version - it flows logically from one point to another and is unusually easy to read for a fossil description!

I have two minor questions/suggestions:

1. The new locomotion section is great, but I notice that the functional analogues for elongate digit 4 all involve fast movement. Based on other characteristics, Termonerpeton is interpreted as relatively heavy with graviportal posture. How do you reconcile these two ideas?
2. The characteristics defining a "fundamentally amniote-like pes" (line 340) appear to be a subset of the "amniote-like characteristics" of the pes (lines 350-350). The first has no citations, but the second has many.

Reviewer #3 (Remarks to the Author):

I thank the authors for providing enhanced images. All my previous comments and points of concern have been adequately addressed. I have no new comments on the revised version of the text.